# Dynamic remodelling of the human host cell proteome and phosphoproteome upon enterovirus infection

Piero Giansanti [1,2,6,10], Jeroen R. P. M. Strating [3,7,10], Kyra A. Y. Defourny [4,10], Ieva Cesonyte [3,10], Alexia M. S. Bottino [3], Harm Post[1,2], Ekaterina G. Viktorova[5], Vien Quang Tri Ho [3,8], Martijn A. Langereis[3,9], George A. Belov [5], Esther N. M. Nolte-'t Hoen[4], Albert J. R. Heck [1,2,10✉] & Frank J. M. van Kuppeveld [3,10✉]

The group of enteroviruses contains many important pathogens for humans, including poliovirus, coxsackievirus, rhinovirus, as well as newly emerging global health threats such as EV-A71 and EV-D68. Here, we describe an unbiased, system-wide and time-resolved analysis of the proteome and phosphoproteome of human cells infected with coxsackievirus B3. Of the ~3,200 proteins quantified throughout the time course, a large amount (~25%) shows a significant change, with the majority being downregulated. We find ~85% of the detected phosphosites to be significantly regulated, implying that most changes occur at the post-translational level. Kinase-motif analysis reveals temporal activation patterns of certain protein kinases, with several CDKs/MAPKs immediately active upon the infection, and basophilic kinases, ATM, and ATR engaging later. Through bioinformatics analysis and dedicated experiments, we identify mTORC1 signalling as a major regulation network during enterovirus infection. We demonstrate that inhibition of mTORC1 activates TFEB, which increases expression of lysosomal and autophagosomal genes, and that TFEB activation facilitates the release of virions in extracellular vesicles via secretory autophagy. Our study provides a rich framework for a system-level understanding of enterovirus-induced perturbations at the protein and signalling pathway levels, forming a base for the development of pharmacological inhibitors to treat enterovirus infections.

[1] Biomolecular Mass Spectrometry and Proteomics, Bijvoet Center for Biomolecular Research and Utrecht Institute for Pharmaceutical Sciences, Utrecht University, Padualaan 8, 3584 CH Utrecht, The Netherlands. [2] Netherlands Proteomics Centre, Padualaan 8, 3584 CH Utrecht, The Netherlands. [3] Virology Section, Division of Infectious Diseases and Immunology, Department of Biomolecular Health Sciences, Faculty of Veterinary Medicine, Utrecht University, Yalelaan 1, 3584 CL Utrecht, The Netherlands. [4] Division of Cell Biology, Metabolism & Cancer, Department of Biomolecular Health Sciences, Faculty of Veterinary Medicine, Utrecht University, Yalelaan 2, 3584 CM Utrecht, The Netherlands. [5] Department of Veterinary Medicine, University of Maryland and VA-MD College of Veterinary Medicine, College Park, MD 20742, USA. [6] Present address: Technical University, Munich, Germany. [7] Present address: Viroclinics Biosciences, Rotterdam, The Netherlands. [8] Present address: Amsterdam University Medical Center, Amsterdam, The Netherlands. [9] Present address: MSD Animal Health, Boxmeer, The Netherlands. [10] These authors contributed equally: Piero Giansanti, Jeroen R. P. M. Strating, Kyra A. Y. Defourny, Ieva Cesonyte, Albert J. R. Heck, Frank J. M. van Kuppeveld. ✉email: a.j.r.heck@uu.nl; f.j.m.vankuppeveld@uu.nl

Viruses are obligate intracellular parasites that reshape the structure, composition and metabolism of host cells to support the infection. In addition, virus infections induce cellular antiviral and stress responses. Here, we investigate the changes that enteroviruses induce in the host cell (phospho) proteome. The life cycle of enteroviruses (reviewed by[1]) is highly conserved and different enteroviruses are thought to modify the host cell in similar ways. Briefly, after receptor-mediated uptake of the naked virions and release of the genome into the cytosol, the viral RNA is translated by the cellular machinery driven by an internal ribosomal entry site in the viral RNA. The viral RNA encodes a single polyprotein that is autoproteolytically processed by the viral proteinases 2A$^{pro}$ and 3C$^{pro}$ into mature capsid proteins VP1 to VP4 and the non-structural proteins 2A–C and 3A–D involved in viral RNA replication.

Enterovirus infection triggers many changes in the ultra-structure and metabolism of the host cell, most of which serve to facilitate virus replication and spreading as well as to suppress cellular antiviral responses[1]. For example, 2A$^{pro}$ cleaves the translation initiation factor eIF4G to suppress cap-dependent translation of host cell mRNAs[2]. The non-structural viral proteins 2BC, 3A and 3D$^{pol}$ remodel cellular membranes into specialised membranous structures (called replication organelles) where 3D$^{pol}$, the viral RNA-dependent RNA polymerase, replicates the viral RNA[1]. Other well-known virus-induced alterations are the defect in nucleo-cytoplasmic trafficking[3] and the disturbance of cytoskeletal proteins and the intermediate filament system. Collectively, these biochemical and structural alterations result in rounding up of cells and detachment from the substrate, a phenomenon known as the cytopathic effect (CPE). Enteroviruses are classically known as lytic viruses, but accumulating evidence points to a role for a non-lytic release mechanism, which may depend on autophagic processes, prior to death and lysis of the host cell[4–6].

Traditionally, studies of enterovirus infection focused on a few selected host proteins. A number of unbiased large-scale approaches at the gene expression level (e.g. siRNA, knockout, and haploid genetic screens) have been performed[7,8]. While in such studies genes can be identified that are important for infection, the regulation and dynamic connections of intracellular signalling pathways remain unresolved. A multitude of protein functions and signalling networks are regulated by rapid and reversible protein phosphorylation[9]. Over the last decades, mass spectrometry (MS) has greatly matured into a robust technique that can now be used to study thousands of proteins and phosphorylation sites in a single experiment. In combination with accurate quantitative methods, proteome and phospho-proteome dynamics can therefore be assessed in an unbiased manner[10], and used to study also host-pathogen interactions[11].

We set out to discover signalling pathways modulated during and/or involved in enterovirus infection, which may uncover new pharmacological approaches for antiviral therapy. To obtain increased insight into virus-induced changes in intracellular signalling in enterovirus-infected cells, we here performed an in-depth system-wide and time-resolved characterization of the changes in the host cell proteome and phosphoproteome of cells infected with coxsackievirus B3 (CVB3), a widely-used model enterovirus. We detected progressive changes in the proteome and phosphoproteome of infected cells, whereby the phosphoproteome underwent the most extensive changes. We dissect phosphorylation events in the mTORC1 signalling pathway and we discover a role for transcription factor EB (TFEB), which was previously not known to be associated with enterovirus infection, in the non-lytic release of enterovirus particles via extracellular vesicles (EV).

## Results and discussion

**Analysis of the proteome of infected cells.** We first set out to investigate the total protein levels in infected cells throughout the course of the enterovirus infectious cycle using CVB3, a model virus commonly used in enterovirus research. Samples from cells infected for different time points until 10 h post infection (hpi), when the replication cycle has been completed and cells display CPE as an indication of their demise, were analysed by single-shot nLC-MS/MS and accurate relative quantification of the proteins was obtained (Fig. 1a). At each time point, between ~26,000 and ~34,000 peptides could be identified, which resulted in the overall identification of 6466 protein groups (Supplementary Data 1) across the 18 samples (six time points, three biological replicates). Of those identified proteins, reliable quantification was obtained for about ~3500 proteins at each time point (Fig. 1b), with a total of 3202 quantified proteins across all the samples (Supplementary Data 1). The 10 h samples had a lower yield, presumably because cells had undergone extensive rounding up and detachment, leading to a relative lower protein yield. Our in-depth quantitative proteome analysis showed very high reproducibility, with a mean Pearson correlation of ~0.96 within biological replicates and of ~0.95 across all the samples (Supplementary Fig. 1a and Source Data File). For an overall assessment of the proteomics similarities and differences of the infected cells, we employed a principal component analysis (PCA), which clearly showed high similarity in the proteome data (Supplementary Fig. 1b and Source Data File). Nevertheless, from 8 hpi substantial changes occurred in the infected cells with the most extensive changes occurring at 10 hpi (Fig. 1c, Supplementary Fig. 1c, and Source Data File).

To satisfy robust statistics, we focused the analysis only on the ~3200 proteins that could be quantified in all samples, thus allowing a detailed analysis of their levels throughout the course of the infection. Of these proteins, 798 (~25%) significantly changed (ANOVA, FDR < 0.05) during infection (Fig. 1d and Source Data File) with a median change in protein level of nearly fourfold (1.96 $^2$log) (Fig. 1e and Source Data File). Unsupervised hierarchical clustering revealed that early time points did not cluster very strongly, indicating that not many changes in the proteome occurred at the early time points. From 6 hpi onward changes in protein abundance became more apparent. Of the differentially regulated proteins 587 proteins decreased in abundance along the progression of the infection, while a second cluster of 211 proteins increased in abundance over time (Fig. 2a, Supplementary Data 1, and Source Data File). While a decrease in protein levels may be explained by the virus-induced shutdown of protein translation, the observation that certain protein levels increased suggests that a subset of cellular proteins escape the bulk translational shutdown induced by viral cleavage of eIF4G. A few groups of proteins stand out that increase during infection (Supplementary Fig. 2a, Supplementary Data 1, and Source Data File), including a subset of mitochondrial proteins and many lysosomal proteins.

We performed a Fisher exact test for all the proteins that significantly changed over time and observed many proteins associated with viral infection, transcription and translation, next to proteins involved in metabolic, membrane, mitochondrial and RNA processes (Fisher exact, FDR < 0.05; Supplementary Data 2, Supplementary Fig. 2b, and Source Data File). The largest clusters of proteins that decreased are linked to the translation machinery, including ribosomal proteins and translation initiation factors (Supplementary Fig. 2b and Source Data File). Many of the mRNAs coding for these proteins possess 5′ terminal oligopyrimidine (5′ TOP) motifs, which allows a regulation of translation independent of bulk mRNAs, e.g. under stress conditions (reviewed in[12]).

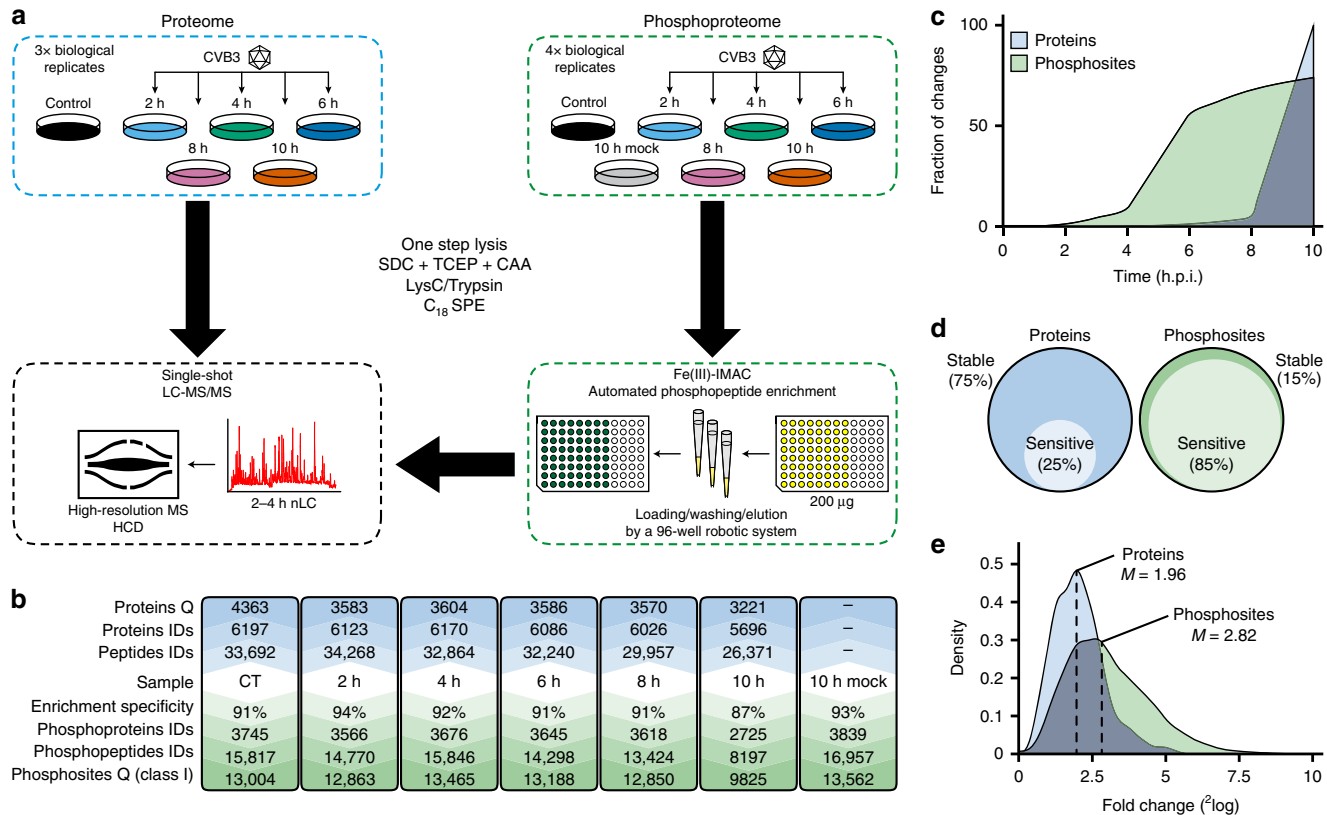

**Fig. 1 Experimental design and overview of the (phospho)proteome data. a** Experimental design for (phospho)proteome analysis of CVB3-infected HeLa cells. Cells were infected with CVB3 for 30 min (MOI 10), medium was replaced and cells were incubated for the indicated amount of time (2–10 h). Control samples were mock infected and incubated for 0 h or 10 h. The colour scheme representing the data of each individual time point is kept consistent throughout the paper. **b** Summary of the identified (IDs) and quantified (Q) peptides, phosphopeptides, proteins, and phosphorylation sites, including enrichment specificity, defined as the fraction of detected phosphopeptides in each sample. **c** Temporal dynamics of global changes in the proteome and phosphoproteome during CVB3 infection, showing that changes in phosphorylation occur much earlier than changes in protein levels. **d** Fraction of proteins and phosphosites dynamically regulated during the infection, as assessed using ANOVA test. **e** Density plot showing the distribution of magnitude of changes of significantly regulated proteins and phosphosites, showing that the differences in phosphorylation are generally more extensive than those at protein levels. Dashed lines indicate the median fold change for each dataset. Source data are provided as a Source Data file for (**c**), (**d**), and (**e**).

We also monitored CVB3 protein levels during the infection, which as expected increased dramatically during infection (Supplementary Fig. 3 and Source Data File). All viral proteins originate from the same precursor, which is proteolytically processed by viral proteases. The specific detection of N-terminal or C-terminal peptides allowed the identification and quantitation of quite a few of these processed viral proteins. Unfortunately, many other viral peptides could not be uniquely ascribed to specific proteins and were assigned to the least processed form in which they occur (Supplementary Data 3). The data on the viral proteins reflects that the infection was highly efficient and reproducible, as at the end of the time course all detected viral proteins were within the top ~40% of detected proteins.

**Analysis of the phosphoproteome of infected cells.** Next, we probed the remodelling of the phosphoproteome throughout the course of infection using a similar workflow as for the full proteome analysis, with the addition of an ultrasensitive and automated phosphopeptide enrichment, which we previously described[13] (Fig. 1a). In most samples, except the 10 h sample, >14,000 phosphopeptides originating from >3500 source proteins could be identified, resulting in the quantification throughout all samples of 16,944 phosphorylation events (Supplementary Data 4), belonging to ~14,000 unique class I phosphorylation sites (Fig. 1b). This high phosphoproteome coverage, even in the

absence of extensive sample pre- or post-fractionation was facilitated by the high enrichment selectivity of the employed Fe (III)-IMAC enrichment strategy, with 87–94% of the identified peptides detected as phosphorylated sequences in each sample (Fig. 1b). A high level of consistency between biological repeats was achieved also in the quantification of the phosphoproteome, with an average Pearson correlation of ~0.88 within biological replicates and of ~0.71 across all the samples (Supplementary Fig. 1a and Source Data File). This slightly lower reproducibility than for the proteome analysis can be attributed to the much higher dynamics and variability of the phosphoproteome. Moreover, the lower correlation measured between time points suggests that substantial changes in phosphorylation occurred in time. This observation was further accentuated by our PCA analysis, which clearly revealed that substantial changes in the phosphorylation levels already occurred in the infected cells at 4 hpi (Supplementary Fig. 1b and Source Data File) and further increased at 6–8 hpi (Fig. 1c, Supplementary Fig. 1c, and Source Data File). Systematic evaluation of differentially regulated proteins and phosphosites (that is, proteins and/or phosphosites that could be quantified in all biological replicates for at least one time point) revealed that about 85% of the quantified phosphoproteome is dynamically regulated over the time course of infection (i.e. the sum of all quantified phosphosites that are changed at any time point divided by the total number of quantified phosphosites), whereas only about a fourth of the quantified proteome

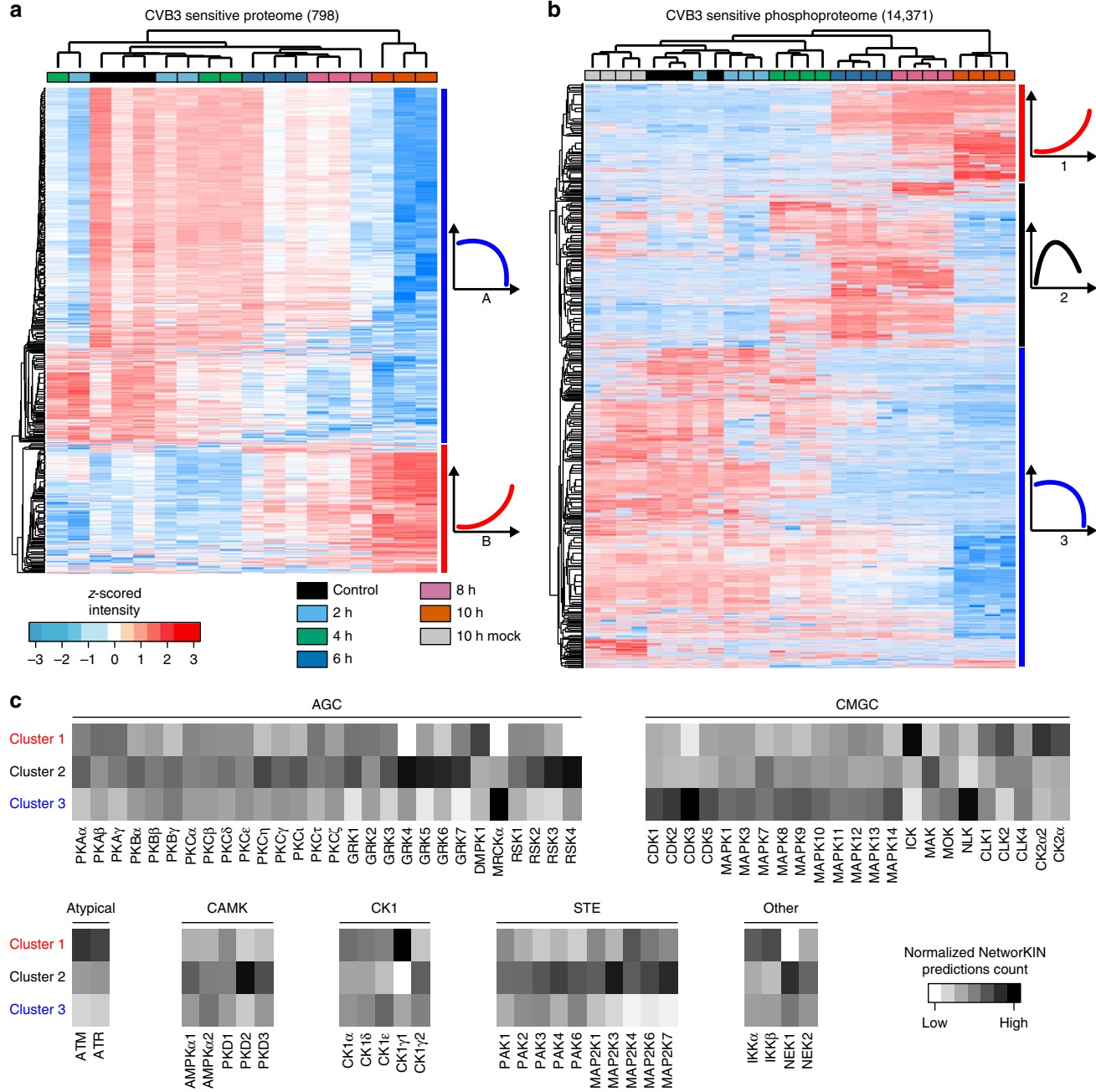

**Fig. 2 Dynamic and functional analysis of the (phospho)proteome data. a** Heat map of z-scored protein abundances (LFQ intensities) of the proteins differentially expressed through time (ANOVA, FDR < 0.05). The two clusters encompass 587 downregulated proteins (cluster A) and 211 upregulated proteins (cluster B). **b** Heat map and hierarchical clustering of the z-scored site intensity of 14,371 CVB3-regulated phosphosites (ANOVA, FDR < 0.05). The three clusters discriminate between upregulated (cluster 1), dynamically regulated (cluster 2) and downregulated (cluster 3) phosphosites. **c** Upstream kinase prediction analysis via NetworKIN of the CVB3-regulated phosphosites suggests that several classes of kinases are activated differently during the progression of the infection. Kinases are grouped based on their families. Source data are provided as a Source Data file for all panels.

undergoes temporal regulation (Fig. 1d and Source Data File). Furthermore, our analysis revealed that protein phosphorylation undergoes the greatest degree of change with a median change of about sevenfold (2.82 $^{2}$log) (Fig. 1e and Source Data File), indicating that the phosphoproteome is much more dynamic and sensitive to CVB3 infection than the proteome.

Our phosphoproteomics analysis cumulatively identified 17,141 class I phosphorylation sites located on 4428 proteins (~2/3 of the total detected proteome), comprising 25,816 serine (86.4%), 2228 threonine (13%) and 97 tyrosine (0.6%) residues (Supplementary Data 4). We performed a hierarchical clustering

analysis of the 14,371 significantly regulated phosphorylation events (ANOVA, FDR < 0.05), defining three main clusters of differentially regulated phosphosites (Fig. 2b and Source Data File). Cluster 1 (2830 sites) contains sites of which the phosphorylation was upregulated relatively late during infection. Cluster 2 (3605 sites) encompassed sites of which the phosphorylation increased early in infection, decreasing again at later times. Cluster 3 (7936 sites) comprises sites of which the phosphorylation decreased during infection. Similar to what the PCA on the entire quantified phosphoproteome revealed, the 10 h mock-infected cells clustered strongly with the other mock

control samples (Supplementary Fig. 1a and Source Data File), indicating that the observed changes in phosphorylation are caused by the effects of the virus infection rather than physiological alterations that occur during the applied incubation time.

To investigate whether specific categories of phosphorylated proteins belong to each of these three broad clusters, we next performed an enrichment analysis (Fisher exact, FDR < 0.05). Specific sets of ontology terms (Supplementary Fig. 2a, Supplementary Data 2, and Source Data File) observed included general categories, such as membrane, cytosol, and cytoskeletal parts and processes, but also specific categories, including Golgi apparatus terms, which were enriched in the late phase of the infection (clusters 1 and 3), and mRNA related processes, which were enriched only in the cluster containing the downregulated sites (cluster 3). Although also several transcription/translation and viral infection-related function groups were clearly affected at the phosphoproteome level (Supplementary Fig. 2a and Source Data File), it appeared that phosphorylation events occurred throughout many different functional groups. Still, when comparing the functional groups that significantly changed in both the proteome and the phosphoproteome, a cluster of transcription & RNA processes and a cluster of translation initiation stood out (Supplementary Fig. 2c and Source Data File).

**Identification of kinases activated by CVB3 infection.** We exploited our extensive phosphoproteomics dataset further and predicted the potential kinases responsible for the above described regulations. We used two independent approaches: (i) motif-x[14], which identifies enriched linear kinase motifs based solely on the amino acid sequence flanking the regulated phosphosites (Supplementary Fig. 4 and Source Data File), and (ii) NetworKIN[15], which predicts upstream kinases responsible for the phosphorylation of the regulated sites (Fig. 2c, Supplementary Data 5, and Source Data File).

Motif-x identified a total of 95 motifs (77 pS and 18 pT), most of them equally represented among the three clusters of regulated phosphosites (centre cloud in the ternary plot in Supplementary Fig. 4 and Source Data File). Those were mainly motifs characterised by the presence of one or multiple acid residues surrounding the phosphorylation site, suggesting casein kinases signalling to be equally active throughout the entire course of the infection. Furthermore, the analysis also revealed several specifically enriched motifs (Supplementary Fig. 4), including (i) the over-representation of pSQ-containing motifs (resembling ATM/ATR substrates) in cluster 1, (ii) the almost exclusive enrichment of motifs with a proline at position +1 (referred to as proline-directed) in cluster 3, and (iii) the enrichment of basophilic-like motifs in cluster 2 and 3. The upstream kinase prediction analysis resulted in a similar picture, allowing us to estimate the possible contribution of each individual kinase to the observed alterations in the phosphoproteome. Accordingly, as expected, targets of ATM/ATR were highly enriched in cluster 1, i.e. gradually increasing over the course of the viral infection, whereas among the basophilic kinases, PKC and several GRKs were mainly predicted for early increase/late decrease sites (cluster 2).

**Signalling pathways affected during CVB3 infection.** While for some phosphosites their effect on the activity of the protein and/ or kinases that phosphorylate these sites are known, the majority of the sites detected here have no annotated function, or are not even reported in large public repositories such as the UNIPROT (www.uniprot.org) or PhosphositePlus (www.phosphosite.org)

databases. Still, analyses of signalling networks can give indications of functional outputs.

To obtain a global picture of the intracellular dynamics during the CVB3 infection, we combined our quantitative proteomics and phosphoproteomics data across all time points. We exploited PhosphoPath software[16] to perform pathway analysis and mapped the regulated proteins and phosphosites on a set of predefined pathways known to be affected by CVB3 or suggested by our data. The analysis revealed that several of the regulated proteins and sites are involved in a multitude of interconnected signalling cascades (Supplementary Fig. 2d and Source Data File). Pathways involved in apoptosis/cell death or transcription/translation were the most enriched ones, followed by PI3K/AKT/mTOR and the MAPK signalling cascades[17–19]. Below, we focused on the mTORC1 pathway, which is known to be inhibited by enteroviruses but for which a detailed and comprehensive oversight is lacking.

**Inactivation of mTORC1 signalling during CVB3 infection.** The mTORC1 signalling pathway has a central role in regulating cellular metabolism, growth, proliferation and survival[20,21]. mTORC1 is a complex of which mTOR is the catalytic subunit. mTORC1 activity is regulated by a vast array of inputs, including growth factors, nutrient availability and cellular energy status. In response to these signals, mTORC1 phosphorylates its targets to control the activity of a range of functional circuits including translation and autophagy (Supplementary Fig. 2e). Our phosphoproteome covered many phosphosites in the mTORC1 network (Fig. 3). Network analysis revealed an overall decrease in mTORC1 kinase activity (Fig. 3). Although, we did not detect the key regulatory residue S2484 on the catalytic subunit mTOR, several phosphorylation events are in line with decreased mTORC1 activity, including a decrease (cluster 3) of PRAS40 S183 autophosphorylation[22,23] and PRAS40 T246 phosphorylation. We also observed events that indicate an increased mTORC1 activity (e.g. increase (cluster 1) of Raptor S863[24,25] and PRAS40 S203[26] phosphorylation), but these occur only very late in infection.

Because mTORC1 activity is hard to deduce from regulatory sites on mTORC1 subunits themselves, we analysed several mTORC1 targets. Phosphoproteomics analysis of eukaryotic translation initiation factor 4E binding protein 1 (4EBP1) revealed a decreased phosphorylation of nearly all detected sites that are directly phosphorylated by mTORC1 (Fig. 3), which we confirmed by Western blotting using several phospho-specific 4EBP1 antibodies (Fig. 4a and Source Data File). Furthermore, 4EBP1 migrates in a gel in multiple bands that represent differentially phosphorylated forms of the protein. In line with a loss of phosphorylation, 4EBP1 gradually collapsed into the lower, least phosphorylated band. Collectively, our data demonstrate a gradual loss of phosphorylation on several residues of 4EBP1 that are known to be directly phosphorylated by mTORC1 from 6 hpi onwards, culminating in a near-complete loss of phosphorylation at 8 hpi. In agreement, we observed reduced (cluster 3) phosphorylation of mTORC1-phosphorylated residues on 4EBP2 and TFEB (see below).

Next, we analysed the ribosomal protein S6 kinase B1 (RPS6KB1, also known as p70$^{S6K}$), which is a hub in controlling mTORC1 downstream signalling towards translation regulation. Although, we did not detect the key regulatory residue S412, which is phosphorylated by mTORC1 and activates p70$^{S6K}$, we could deduce a clear inhibition of p70$^{S6K}$ activity from the reduced (cluster 3) phosphorylation of six out of seven detected p70$^{S6K}$ target sites on ribosomal protein S6 (RPS6), programmed cell death protein 4 (PDCD4) and eukaryotic initiation factor 4B (eIF4B) (Fig. 3), which collectively lead to a reduced translation

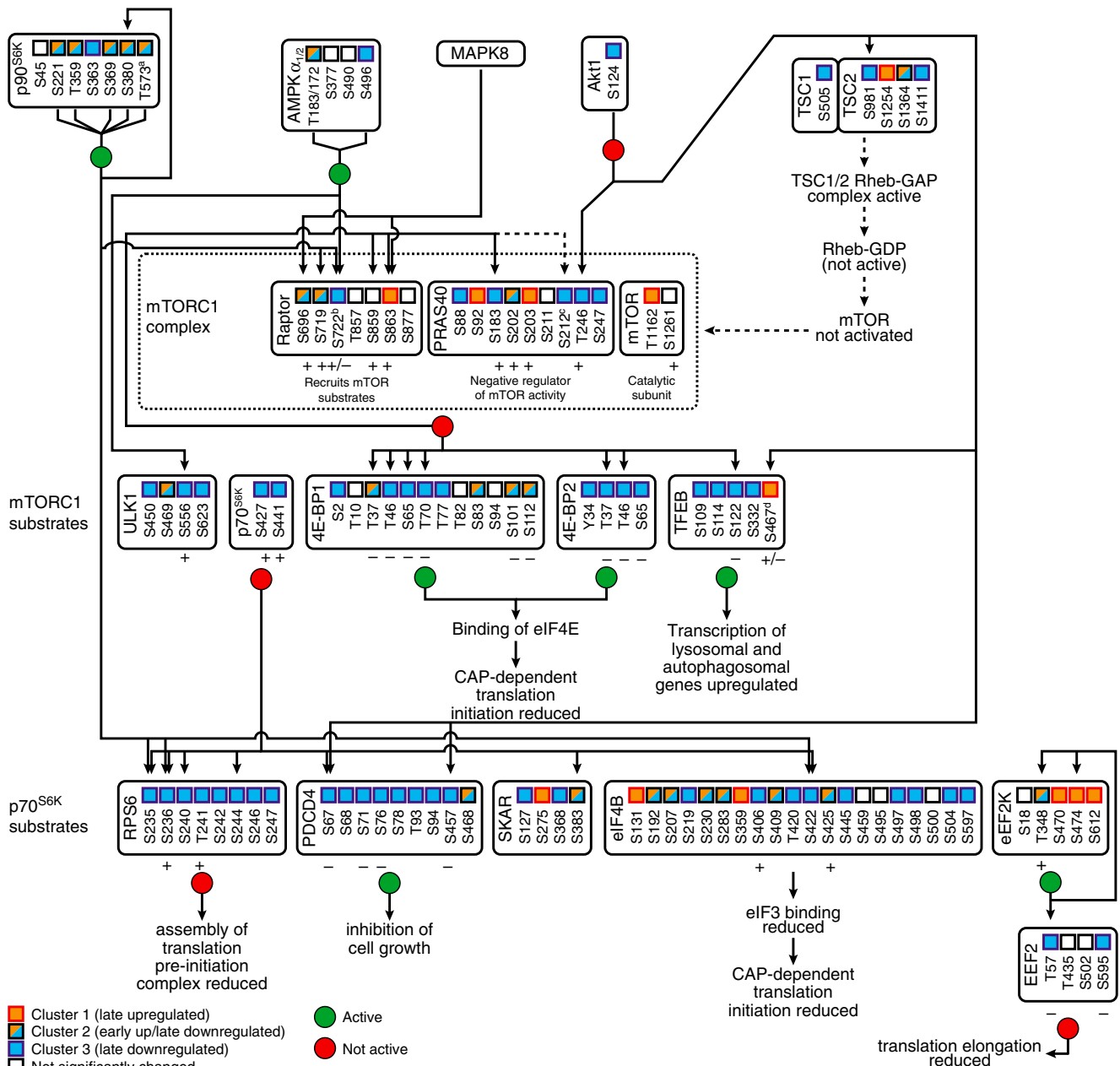

**Fig. 3 Overview of phosphorylation events in the mTORC1 signalling in CVB3-infected cells.** Analysis of phosphosites on proteins in the mTORC1 signalling network and some upstream kinases, restricted to these kinases of which sites on the mTORC1 complex are detected. Supplementary Data 4 list the detected phosphosites and the clusters to which significantly changed sites belong. For a detailed description, see the main text. The effect of the phosphorylation on protein activity is denoted with plus sign and minus sign, indicating increase and decrease, respectively. In case a site has multiple phosphorylation forms, the pattern for the lowest detected phosphorylation multiplicity is shown.

and cell growth. We confirmed this by Western blotting with antibodies that specifically recognise RPS6 phosphorylated at S240/244 (Fig. 4a and Source Data File). A reduced p70[S6K] activity would also lead to a decrease in phosphorylation of the inhibitory site S366 on eukaryotic elongation factor 2 kinase (eEF2K). Although we did not detect this site, we could deduce an increased eEF2K activity from the increased phosphorylation on eEF2K T348 (early increase) and S474 (late increase), which are autocatalytically phosphorylated[27,28]. eEF2 T57 (in literature often referred to as T56) is directly phosphorylated by eEF2K and inhibits the function of eEF2 in translation elongation[29]. Indeed, the phosphoproteome data showed that eEF2 p-T57 levels increased at early time points, although levels decreased late

in infection (cluster 3) (Supplementary Data 4). Western blotting confirmed that this site is phosphorylated during infection. Only at 10 hpi the phosphorylation level dropped, which is likely caused by the decrease in total eEF2 levels, as detected in the full proteome analysis (log2 fold change −1.84, Supplementary Data 1) and by Western blotting (Fig. 4a and Source Data File).

Altogether, our data are in line with an inhibition of mTORC1 activity leading to a repression of cap-dependent host mRNA translation. Enteroviruses also repress cap-dependent initiation of host mRNA translation by cleaving eukaryotic translation initiation factor 4G (eIF4G). Inhibition of mTORC1 may suppress translation of subsets of proteins that are insensitive to

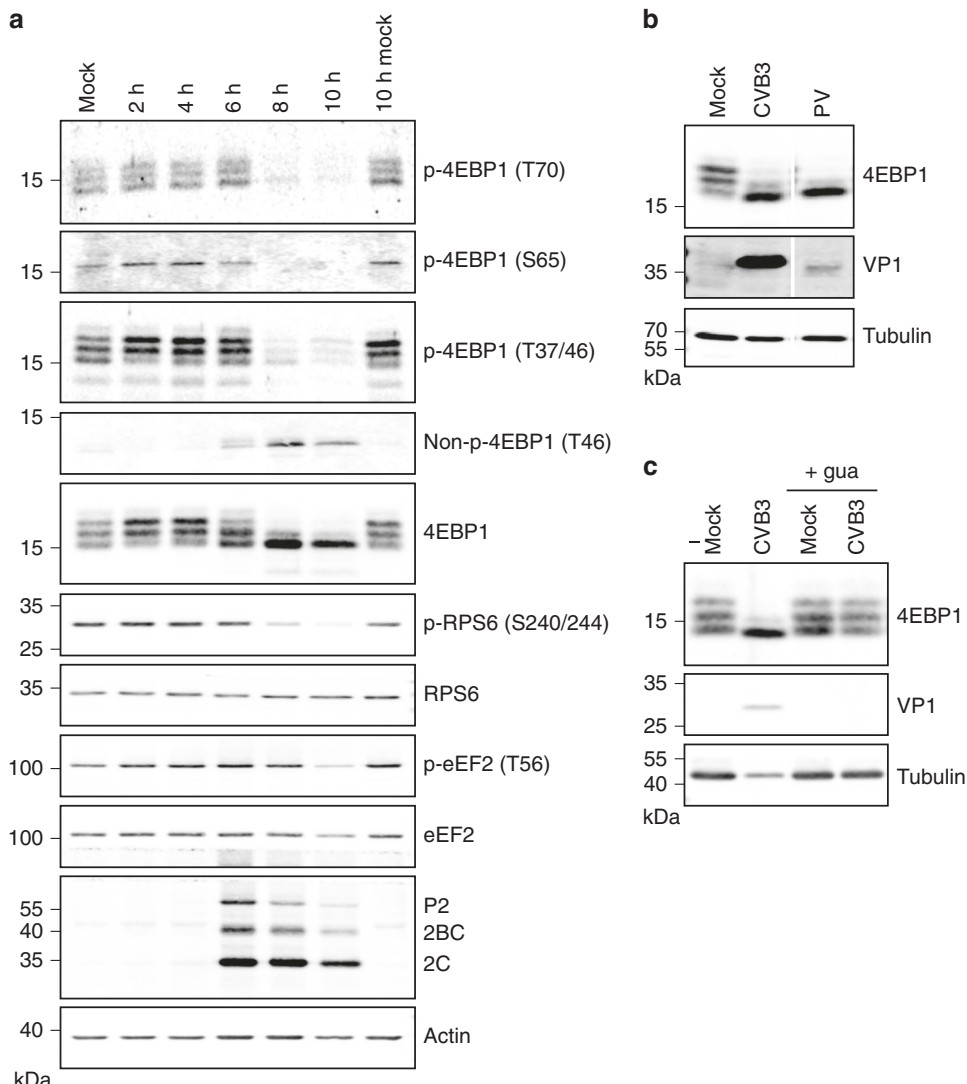

**Fig. 4 Analysis of the activity of mTORC1 in enterovirus-infected cells. a** Western blot analysis of infected cells showing mTORC1 inhibition. HeLa cells were infected with CVB3 at MOI 50 as in Fig. 1a, lysed at different time points and the lysates were analysed by Western blotting using antibodies that recognise proteins in the mTORC1 signalling pathway or specific phosphorylated forms of these proteins. The different 4EBP1 bands represent differentially phosphorylated forms of the protein, whereby slower migrating bands represent more extensively phosphorylated forms of the protein. An antibody against viral protein 2C, which also detects the 2C cleavage precursor proteins 2BC and P2, serves as a control to show infection. Actin is used as a loading control. The 10 h infected samples contain less material due to cytopathic effect and the ensuing detachment of part of the infected cells. **b** HeLa cells were infected with CVB3 (enterovirus B species) or poliovirus (PV, enterovirus C species) at MOI 25 and Western blot analysis of 4EBP was performed as in panel (**a**). An antibody against the capsid protein VP1 of CVB5, which is highly cross-reactive to CVB3 VP1 and to a lesser extent also recognises PV VP1, was used as an infection control. Tubulin was used as a loading control. **c** HeLa cells were infected with CVB3, treated with 2 mM of the replication inhibitor guanidine as indicated, lysed after 8 h, and processed for Western blot analysis as in (**a**). mTORC1 inhibition is evident from a collapse of the 4EBP1 bands into the least phosphorylated lower band only when cells were infected with CVB3 in the absence of guanidine. Source data are provided as a Source Data file for all panels.

eIF4G cleavage, and thereby the two pathways likely complement each other to achieve global host mRNA translation shut-off. Intriguingly, we observe that a part of the proteome is upregulated during infection, implying a different level of regulation.

**Multiple pathways contribute to mTORC1 regulation during infection.** mTORC1 integrates input signals from many different signalling pathways[20]. Analysis of phosphorylation events on several upstream regulators of mTORC1 revealed the involvement of several different regulators (Akt1-TSC1/2, p90[S6K], AMPK and MAPK8) in mTORC1 inhibition (Fig. 3). Signalling via the Akt1-TSC1/2-Rheb axis is the canonical mTORC1 regulatory pathway in

response to extracellular signals like insulin[20]. We observed a decreased Akt1 S124 phosphorylation, which indicates a reduced Akt1 activity[30,31]. Correspondingly, phosphorylation of the downstream residue S981 of TSC2 was reduced, which indirectly correlates with reduced mTORC1 activity[32]. Likewise, PRAS40 T246 phosphorylation—which relieves the inhibitory activity of PRAS40 on mTORC1 activity[33–36]—was reduced (cluster 3), in line with reduced mTORC1 activity.

We also detected signalling of the kinases mitogen-activated protein kinase 8 (MAPK8), AMPK and ribosomal protein S6 kinase A1 (RPS6KA1, also known as p90[S6K]) culminating on the mTORC1 subunit Raptor, which recruits mTORC1 substrates[21]. Phosphorylation of the MAPK8-downstream sites S696 (cluster

2) and S863 (cluster 1) on Raptor was (transiently) increased, which have been reported to correlate with enhanced mTORC1 activity[24,25,37]. Hence, MAPK8 is unlikely inhibited in infected cells. The activity of p90S6K is also unlikely inhibited. We observed transient upregulation (cluster 2) of several phosphorylation events on p90S6K that are associated with activation, including S221, T359, T573 and S380, the latter of which is autocatalytically phosphorylated[38], although phosphorylation of S363, which is also associated with activation[38], was down-regulated (cluster 3).

AMPK is a pivotal sensor of the cellular energy status that contributes to mTORC1 inhibition through Raptor S722 phosphorylation[39]. We observed several phosphorylation events on AMPK that imply an early AMPK activation and a consistent early increase in phosphorylation of the AMPK target residue GBF1 T1337 (discussed above). Yet, we observed a decreased (cluster 3) phosphorylation of the AMPK target residue Raptor S722 and phosphorylation of the AMPK target ULK1 S556 was also downregulated. Hence, a direct correlation between phosphorylation of regulatory sites on AMPK and signalling output of the mTORC1 network is difficult to establish. It is important to stress that mTORC1 activity is delicately tuned by many different parameters, that many mTORC1 substrates are also phosphorylated by other proteins and that different mTORC1 substrates may be differentially regulated and compete with each other[22,40].

**mTORC1 inactivation relies on CVB3 translation and replication.** The data described above clearly demonstrate mTORC1 inactivation by CVB3. This inactivation has been reported for other enteroviruses, including poliovirus[41] (Fig. 4b and Source Data File), and was observed in all cell lines tested (Supplementary Fig. 5a and Source Data File). Many different events, induced during different stages of the viral life cycle, can influence mTORC1 activity. We hypothesised that cellular signalling events induced during CVB3 entry sufficed to induce mTORC1 inhibition. Enterovirus entry involves delivery of the viral genome over the lysosomal limiting membrane, which requires the membrane to be punctured in a yet unknown way. This leads to a recruitment of machinery to signal and repair the damage, which include the phospholipase A2 G16 (PLA2G16) and galectin-8[42]. Several galectins, including galectin-8, were recently reported to be involved in regulating mTORC1 activity[43]. To test whether lysosomal damage occurring during CVB3 entry sufficed to induce the observed inhibition of mTORC1, we infected cells with CVB3 and incubated them for 8 h with the replication inhibitor guanidine. Under this condition, CVB3 infection did not lead to an inhibition of mTORC1 activity, as evident from an unaltered 4EBP1 pattern (Fig. 4c and Source Data File). Hence, the inhibition of mTORC1 activity during enterovirus infection depends on viral genome replication and/or the ensuing high viral protein expression levels. Notably, mTORC1 inhibition is not merely a consequence of the accumulation of picornavirus dsRNA, as it does not occur in cells infected with encephalomyocarditis virus (EMCV), a picornavirus belonging to the genus *Cardiovirus* (Supplementary Fig. 5b and Source Data File).

**mTORC1 downstream transcription factor EB (TFEB) affects non-lytic virus release via extracellular vesicles.** Autophagy is induced upon enterovirus infection and has been suggested to be involved in various stages of the viral life cycle, including viral RNA replication, virion assembly and release[4–6]. Autophagy induction upon enterovirus infection involves activation of ULK1, a key inducer of autophagy that is repressed by mTORC1. In addition, mTORC1 controls the transcription of genes encoding proteins functioning in autophagosomes and lysosomes through

repressive phosphorylation events on several key residues of the TFEB[44] (reviewed in[45]). While we did not detect mTORC1-dependent phosphorylations on ULK1 during infection, we detected decreased phosphorylation of a previously reported mTORC1-phosphorylated inhibitory site on TFEB (S122)[46] (Fig. 3). Correspondingly, we observed increased RNA levels of TFEB-regulated genes following infection (Supplementary Fig. 5c and Source Data File), together with increased lysosomal proteins levels seen in the proteomics experiment and by Western blotting (Supplementary Fig. 2b, Supplementary Fig. 5e, Supplementary Fig. 6, and Source Data File).

Given the intricate relationship between autophagy and the viral life cycle, we investigated whether TFEB is a key factor in enterovirus infection using TFEB knockout (TFEB[KO]) cells. In these cells we confirmed a causal link between TFEB activation and increased lysosomal/autophagosomal gene expression (Supplementary Fig. S5f and Source Data File). Using a luciferase-expressing reporter virus, we observed no effect of TFEB knockout on virus replication (Supplementary Fig. 5g and Source Data File). Similar luciferase levels in the presence of the replication inhibitor guanidine in wildtype and TFEB[KO] cells indicated that also translation of the viral polyprotein is not affected by TFEB knockout (Supplementary Fig. 5g and Source Data File).

While the intracellular virus levels remained unchanged, we consistently observed a 5- to 10-fold reduction of extracellular viral titers at 8 hpi in TFEB[KO] cells (Fig. 5a and Source Data File). The difference in extracellular virus was not caused by differences in cellular integrity, as infected cells at 8 hpi were still in the early stages of the (rapid) induction of cell death and the amount of cell lysis was similar in infected wildtype and TFEB[KO] cells (Fig. 5b, Supplementary Fig. 7a, and Source Data File). In addition to the induction of cell lysis, viruses can also be released non-lytically from the infected host cell, most notably via the budding and release of virions inside lipid-bilayer enclosed EV. The autophagic pathway has previously been implicated in the release of EV-enclosed enteroviruses based on detection of the autophagy marker LC3 on EV released during infection and on the observed link between autophagy levels and non-lytic virus release[47–50]. Therefore, we assessed whether TFEB knockout affected non-lytic virus release via EV. EV were isolated from the supernatant of infected cells by pelleting at $100,000 \times g$. Due to their high lipid content, EV have a lower buoyant density than naked virions. This feature can be used to separate EV-enclosed viruses from co-isolated naked CVB3 virions on a density gradient as was confirmed by triton-X100 treatment (Supplementary Fig. 7b and Source Data File). Quantification of the virus recovery within naked virus and EV-containing density fractions showed a significant decrease in EV-enclosed virus release in the absence of TFEB (Fig. 5c, Supplementary Fig. 7c, and Source Data File). In addition, a trend for a reduction in the release of naked virus particles was observed. This reduction may be due to variability in the experiments although we cannot formally rule out a contribution of TFEB in lytic virus release. Since TFEB is a central hub within the autophagic machinery, the ability of CVB3 to induce autophagy in wildtype and TFEB[KO] cells was monitored by measuring LC3 lipidation by Western blotting. While no defects in intracellular autophagy levels were observed in either mock-treated or infected TFEB[KO] cells (Fig. 5d and Source Data File), TFEB knockout almost entirely prevented the secretion of LC3 within EV isolates from infected cells (Fig. 5e and Source Data File). This result corroborates the observed decrease in EV-enclosed virus release (Fig. 5c and Source Data File). TFEB thus appears to potentiate a secretory autophagic pathway that is switched on during infection to mediate non-lytic virus release. This TFEB-dependent release of EV-enclosed virus

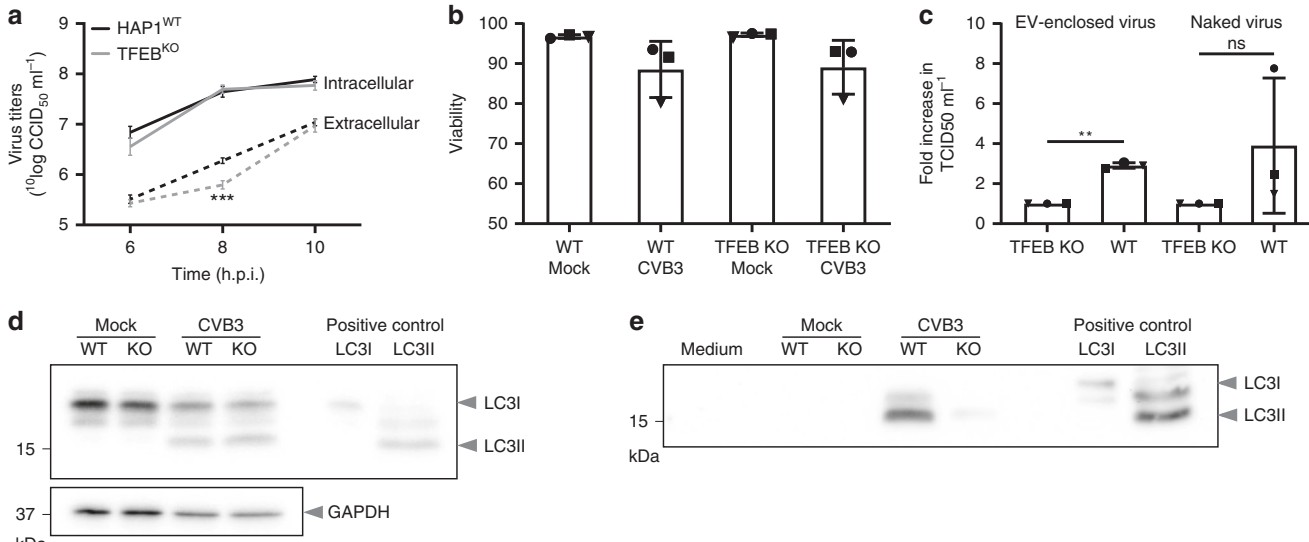

**Fig. 5 TFEB promotes non-lytic release of CVB3 in early stages of infection. a** Accumulation of extracellular virus is delayed in cells lacking TFEB. Wild-type (HAP1$^{WT}$) and TFEB knockout (TFEB$^{KO}$) HAP1 cells were infected with CVB3 at MOI 1 as in Fig. 4a. Intracellular and extracellular virus titers at the indicated time points were determined by end-point titration. Shown are means, error bars represent ± SEM ($n = 3$ independent experiment, each with biological triplicates; results of all replicates were averaged). Results were statistically evaluated by two-way ANOVA with Bonferroni's post-hoc test. ***, $p < 0.001$ ($p = 0.0005$). Plots showing the individual data points are available in the Source data file. **b** Cell viability at 8 hpi is not affected by TFEB knockout. HAP1$^{WT}$ or TFEB$^{KO}$ HAP1 cells were infected with CVB3 at MOI 10. Cell viability at 8 hpi was determined using a FACS-based viability staining. Shown are means, error bars represent ± SD. Different symbols indicate data of three independent experiments. **c** TFEB knockout impairs extracellular vesicle-enclosed virus release. Virus released in the supernatant at 8 hpi was isolated by pelleting at 100,000 × $g$ and purified using density gradient centrifugation. Infectivity retrieved within naked virus (1.35–1.15 g ml$^{-1}$) and EV-enclosed virus containing fractions (1.10–1.04 g ml$^{-1}$) was determined using end-point titration. The ratio between virus recovery from samples from HAP1$^{WT}$ and TFEB$^{KO}$ HAP1 cells is depicted after correction for any difference in intracellular titers. Shown are means, error bars represent ± SD ($n = 3$ independent experiments). Results were statistically evaluated using a one-sample $t$-test. Details on the statistics are included in the Source data file. **, $p < 0.005$ ($p = 0.0017$). **d, e** TFEB knockout affects the induction of secretory autophagy. Western blot analysis of the autophagy-related protein LC3 in whole cell lysates (**d**) and 100,000 × $g$ pelleted material from cell culture supernatants (**e**) from CVB3-infected HAP1$^{WT}$ or TFEB$^{KO}$ HAP1 cells at 8 hpi. Representative image of three independent experiments are shown. Source data are provided as a Source Data file for all panels.

occurred only during a narrow time frame (Fig. 5a and Source Data File). An explanation for this finding is that prior to the 8 hpi time point no infectious virus has been formed while later in infection lytic release is the major source of extracellular virus.

**Phosphorylation of host factors involved in formation of viral replication organelles.** Genome replication of enteroviruses is associated with specialised membranous structures (called replication organelles), which are presumably generated by remodelling the endoplasmic reticulum and/or Golgi. Enteroviruses usurp a specific set of cellular proteins to build replication organelles, including the Golgi-derived proteins Golgi-specific brefeldin A-sensitive factor 1 (GBF1), acyl-CoA binding domain-containing protein 3 (ACBD3), phosphatidylinositol 4-kinase type IIIβ (PI4KB) and oxysterol-binding protein (OSBP) (reviewed in[1]). We detected phosphosites on GBF1, PI4KB and OSBP, but not ACBD3, of which some have previously been linked to Golgi integrity (Supplementary Fig. 8a) and hence may play a role in replication organelle formation. We focused on the phosphosites GBF1 T1337 and PI4KB S266 because these sites have been linked to Golgi integrity and because for these proteins suitable experimental models were available to test the roles of the sites in enterovirus replication. As discussed more extensively in the Supplementary text and shown in Supplementary Fig. 8, when the activity or expression of the endogenous proteins were disrupted, phosphomimetic and non-phosphorylatable mutants of these sites could both restore virus replication, indicating that phosphorylation at sites that have been implicated to be important for membrane homoeostasis in uninfected cells are not important for

enteroviruses to build their replication organelles. These data lend further support to the idea that enteroviruses can rewire cellular pathways for efficient replication.

Our unbiased proteome and phosphoproteome analysis of enterovirus infected human host cells revealed that infection-induced changes in the cellular proteome occur at multiple levels, although the most dramatic changes occur at the post-translational level. The attribution of functional effects is severely hampered by the enormous lack of functional knowledge for most phosphosites. Still, network analysis and dedicated experiments allowed us to dissect mTORC1 signalling as a major network of which the signalling is changed during virus infection, which via TFEB may contribute to virus spread via EV. Our (phospho) proteome datasets provide a rich resource for further analysis of enterovirus-induced changes in the cellular infrastructure and metabolism. This is essential for better understanding viral pathogenesis and may open new possibilities for development of pharmacological inhibitors to treat enterovirus infections.

## Methods

**Cell culture and virus infection.** HeLa R19 cells (obtained from George Belov, University of Maryland, College Park (MD), USA), BGM (ATCC), Vero E6 (ATCC), HuH7 (ATCC) and A549 (National Institute of Public Health and the Environment (RIVM), Bilthoven, The Netherlands) cells were cultured in DMEM (Lonza) supplemented with 10% foetal calf serum (FCS) and penicillin/strepto-mycin. HAP1 TFEB$^{KO}$ cells (Horizon Discoveries #HZGHC003476c010) and HAP1 parental cells (Horizon Discoveries C631) were cultured in IMDM (Thermo Fisher Scientific) supplemented with 10% FCS and penicillin/streptomycin. Cells were cultured at 37 °C at 5% CO$_2$. Knockout of the TFEB gene was confirmed by sequencing of the cellular genomic DNA using the supplier's protocol.

Viruses used were CVB3 strain Nancy, poliovirus 1 strain Sabin and EMCV strain Mengovirus. RLuc-CVB3 was generated from RNA as described in[51].

For virus infection, medium was aspirated and virus dilution was added. For mock infection, medium without virus was added. Cells were infected at the indicated multiplicity of infection (MOI) for 30 min at 37 °C and 5% CO$_2$. Virus dilution was aspirated and fresh medium was added. In some cases, 2 mM of the replication inhibitor guanidine was added. Infected cells were further incubated at 37 °C and 5% CO$_2$ for the indicated times.

For experiments with *Renilla* luciferase-expressing reporter virus, medium was aspirated after the incubation time, cells were lysed and luciferase activity was determined using the *Renilla* luciferase assay (Promega). To determine intracellular and extracellular titers, the medium—which contains extracellular virus—was transferred to a fresh plate at the end of the incubation time. Fresh medium was added to the cell layer, which contains the intracellular virus. Both plates were frozen at −20 °C and freeze-thawed three times to disrupt cells and release all virus. Virus titers were determined by end-point dilution on HeLa R19 cells and calculated by the method of Reed and Muench.

**Protein lysis and digestion for (phospho)proteomics analysis**. Medium was aspirated, cells were washed with ice-cold PBS and incubated with ice-cold TEN buffer (40 mM Tris-HCl pH7.4, 10 mM EDTA, 150 mM NaCl) on ice for 5 min. Cells were released from the plates by pipetting with a P1000 pipette and collected in 1.5 ml vials. Cells were pelleted by centrifugation for 5 min at 500 × *g* at 4 °C. Cell pellets were resuspended in PBS, cells were pelleted again and pellets were snap-frozen in liquid nitrogen and stored overnight at −80 °C until further analysis. Pellets were thawed, cells were lysed, reduced, and alkylated in lysis buffer containing 1% sodium deoxycholate (SDC), 10 mM tris(2-carboxyethyl)phosphinehydrochloride (TCEP), 40 mM chloroacetamide (CAA), and 100 mM TRIS, pH 8.0 supplemented with complete EDTA-free protease inhibitor mixture (Roche), and phosSTOP phosphatase inhibitor mixture (Roche) for the phosphoproteomics analysis. Cells were heated for 5 min at 95 °C, sonicated with a Bioruptor Plus (Diagenode) for 15 cycles of 30 s, and diluted 1:10 with 50 mM ammonium bicarbonate, pH 8.0. The total protein concentration was determined using a Bradford assay (Bio-Rad). Proteins were digested overnight at 37 °C with trypsin (Promega) with an enzyme:substrate ratio of 1:50 and Lys-C (Wako) with an enzyme:substrate ratio of 1:75[52]. SDC was precipitated by acidification to 5% of formic acid (FA). Samples were desalted using Sep-Pak C18 cartridges (Waters), eluted with 80% acetonitrile (ACN)/0.1% trifluoroacetic acid (TFA) and directly subjected to phosphopeptide enrichment. Samples for proteome analysis were instead dried down and stored at −80 °C until subjected to nLC-MS analysis.

**Automated Fe(III)-IMAC phosphopeptide enrichment**. Phosphorylated peptides were enriched using Fe(III)-NTA in an automated fashion[13] using the AssayMAP Bravo Platform (both Agilent Technologies). Fe(III)-NTA cartridges were washed with 250 μL of 100% ACN/0.1% TFA and conditioned using 250 μL of loading buffer consisting of 80% ACN/0.1% TFA. Samples were dissolved in 200 μL of loading buffer and loaded onto the cartridge. Subsequently, the columns were washed with 250 μL of loading buffer, and then the phosphopeptides were eluted with 25 μL of 1% ammonia directly into 25 μL of 10% FA in water. Samples were dried down and stored at −80 °C until subjected to nLC-MS analysis.

**Reverse phase chromatography and mass spectrometry**. Peptides were subjected to reversed phase nLC-MS/MS analysis using an Agilent 1290 Infinity UHPLC system (Agilent) coupled to an Orbitrap Q Exactive Plus mass spectrometer, or Orbitrap Fusion mass spectrometer (both Thermo Scientific) for the phosphoproteome analysis. The UHPLC was equipped with a double frit trapping column (Dr Maisch Reprosil C18, 3 μm, 2 cm × 100 μm) and a single frit analytical column (Agilent Poroshell EC-C18, 2.7 μm, 50 cm × 75 μm). Trapping was performed for 5 min in solvent A (0.1% FA in water) at 5 μl min$^{-1}$, while for the elution the flow rate was passively split[53] to 300 nl min$^{-1}$. The gradient was as follows: 13–40% solvent B (0.1% FA in 80% ACN) in 220 min (or 8–32% in 95 min for phosphopeptide analysis), to 100% solvent B in 3 min, 1 min of 100% solvent B, and finally equilibration of the chromatographic columns with 100% solvent A for the following 10 min before injection of the next sample. Total analysis time was 235 min for the proteome samples and 110 min for the phosphoproteome samples.

The mass spectrometer was operated via Xcalibur (version 3.0.63, Thermo Scientific) in data-dependent mode. The Orbitrap Q Exactive Plus full-scan MS spectra from m/z 375–1600 were acquired at a resolution of 35000 (FWHM) after accumulation to a target value of 3e$^6$. Up to ten most intense precursor ions were selected for fragmentation, with the isolation window set to 1.5 m/z. HCD fragmentation was performed at normalised collision energy of 25% after the accumulation to a target value of 5e$^4$. MS/MS was acquired at a resolution of 17,500 (FWHM). Dynamic exclusion was set to 36 s. The Orbitrap Fusion full-scan MS spectra from m/z 375–1500 were acquired at a resolution of 120,000 (FWHM) after accumulation to a target value of 4e$^5$. The most intense peptide ions fitting within a 3 s cycle were selected for HCD fragmentation, with the isolation window set to 1.6 m/z, and a normalised collision energy of 30%, after the accumulation to a target value of 5e$^4$. MS/MS was acquired at a resolution of 30,000 (FWHM). Dynamic exclusion was set to 16 s.

**Data analysis of (phospho)proteomics**. Raw files were processed using Max-Quant (version 1.5.8.0)[54]. Proteins and peptides were identified using a target-decoy approach with a reversed database, using the Andromeda search engine integrated into the MaxQuant environment. The database search was performed against the human Swiss-Prot database (version November, 2015—20,193 entries) supplemented with CVB3 proteins sequences (21 entries), and against a common contaminants database (245 entries). Default settings were used, with the following minor changes: methionine oxidation, protein N-term acetylation, and phosphorylation of serine, threonine, and tyrosine as variable modifications. Enzyme specificity was set to trypsin with a maximum of two missed cleavages and a minimum peptide length of seven amino acids. A false discovery rate (FDR) of 1% was applied at the protein, peptide, and modification level. A site localisation probability of at least 0.75 was used as thresholds for the localisation of phosphorylated residues (class I phosphosites). Label-free quantification via MaxLFQ algorithm[55] was performed, and "match between runs" was enabled. For each phosphorylation site, MaxQuant provides three quantification values (multiplicity): one when the site is quantified on a singly phosphorylated peptide, one when quantified on a doubly phosphorylated peptide and, one when quantified on a triply (or more) phosphorylated peptide. In this paper, we refer to these three values as phosphorylation events.

Bioinformatics analysis was performed with Perseus[56], Microsoft Excel and R statistical computing software[57]. R and GraphPad (version 5 and 7) were used to plot graphs. All figures were finally organised in Adobe Illustrator CS6.

Prior to statistical analysis, both proteome and phosphoproteome datasets were normalised by median normalisation, filtered to retain only proteins or sites that have been quantified in at least three biological replicates in at least one experimental condition (i.e. time point), and missing values were imputed in Perseus using default settings. Significance was assessed by ANOVA, whereas a parametric two-tailed Welch's *t*-test with a permutation-based FDR of 5% and a S0 parameter of 1 was used to evaluate the temporal dynamics and magnitude of regulations, when comparing infected cells versus the control sample at 0 h.

For hierarchical clustering, logarithmized LFQ intensities of significantly regulated proteins or phosphosites were first z-scored and clustered using Euclidean as a distance measure for column and row clustering. The number of clusters was selected by manual inspection. Annotations were downloaded within Perseus and extracted from UniProtKB, Gene Ontology (GO), and Kyoto Encyclopedia of Genes and Genomes (KEGG). Reactome pathways were obtained by PANTHER (http://pantherdb.org). Enrichment scores were determined using Fisher exact test, either in Perseus or the R package topGO.

Significantly regulated phosphorylation sites were subjected to motif-x analysis[14], using the following parameters: minimum occurrence of 20, significance threshold of 1e$^{-6}$ and the entire human proteome (Swiss-Prot database, see above) as background reference set. Analysis was performed using the R package 'rmotifx'[58]. Upstream kinases responsible for the observed phosphorylation sites were also predicted by using the NetworKIN algorithm[59]. A score threshold of 1 was applied, and only the top 5 predictions were used. The normalised prediction count for a given kinase was calculated as follows: i. the amount of predictions of that kinase in a given cluster divided by the total prediction counts obtained in that cluster and then ii. the obtained value was normalised to 100% on the sum of the three values obtained from each cluster.

Signalling networks were visualised in Cytoscape using the PhosphoPath plugin[16]. Here, the list of differential expressed entries was imported in PhosphoPath and mapped against data from PhosphoSitePlus[60], Biogrid[61], and Wikipathways[62] for the generation of a protein–protein interaction network and pathway analysis. The entire human proteome was used as background to calculate pathway enrichment (Fisher exact test) of the regulated sites/proteins.

**EV isolation**. HAP1$^{WT}$ or TFEB$^{KO}$ cells were infected at MOI 10 for 1 h at 37 °C and 5% CO$_2$. Cells were washed three times with PBS (+Ca+Mg) and supplemented with fresh medium. After 7 h of incubation (8 h.p.i.) the EV-containing medium was collected. Cells and debris were removed by differential centrifugation for 10 min at 200 × *g* and 2 × 10 min at 500 × *g*. Next, EV were pelleted using ultracentrifugation for 65 min at 28,000 rpm in an SW40 rotor (κ-factor 280.3) (Beckman Coulter, Brea, CA). The resulting pellet was resuspended in 2× Laemmli sample buffer (reducing) for immunoblotting or in 0.1% BSA/PBS (precleared from aggregates by centrifugation for 16 h at 100,000 × *g*) for further purification on a density gradient. For density gradient purification, samples were mixed with 0.1% triton or left untreated for 15 min and subsequently mixed with 60% w/v iodixanol (Optiprep, Axis-Shield, Oslo, Norway) to a final concentration of 45%. Samples were then overlaid with eight equal volume layers of iodixanol ranging from 40 to 5% w/v. EV were floated upwards into the gradient by centrifugation for 16 h at 39,000 rpm in an SW40 rotor (κ-factor 144.5). 1 ml fractions were collected from the top of the gradient and their density was determined using a refractometer.

**Determining naked versus EV-enclosed virus release**. Virus titers in gradient fractions with densities 1.35–1.15 g ml$^{-1}$ (containing naked virus) and 1.10–1.04 g ml$^{-1}$ (containing EV-enclosed virus) were analysed by end-point dilution on HeLa R19 using the method of Spearman–Karber. Differences in extracellular virus release by HAP1$^{WT}$ versus TFEB$^{KO}$ cells were corrected for difference in intracellular virus titers. Intracellular virus titers were determined using the following

procedure: cells were harvested using trypsin and pooled with any detached cells isolated during the $200 \times g$ preclearing step of the EV isolation protocol. Samples were freeze-thawed three times, cleared by centrifugation for 10 min at $500 \times g$ and analysed by end-point dilution assay.

**Cell viability assay**. To assess cell viability, cells were harvested 8 h.p.i. as described above for intracellular virus titration. Cells were washed twice in cold PBS and stained for 30 min on ice using Fixable Viability Dye eFluor 506 (eBioscience, San Diego, CA) according to the manufacturer's protocol. Unbound dye was removed by washing with PBS and the cells were fixed in 1% paraformaldehyde. After fixation samples were washed in 1% BSA in PBS and analysed using a CytoFLEX LX (Beckman Coulter). Data were analysed using FlowJo v10.07 software (FlowJo, Ashland, OR, USA).

**Immunoblotting**. A description of the experimental procedures is in the supplementary information.

**Reporting summary**. Further information on research design is available in the Nature Research Reporting Summary linked to this article.

## Data availability

The MS proteomics data have been deposited in the ProteomeXchange Consortium via the PRIDE partner repository[63] with the dataset identifier PXD011163. Annotation from UniprotKB, Gene Onthology (GO), and Kyoto Encyclopedia of Genes and Genomes (KEGG) were downloaded with Perseus. Reactome pathways were obtained by PANTHER (http://pantherdb.org). PhosphoSitePlus, Biogrid, and Wikipathways databases are embedded or directly queried by Cytoscape/PhosphoPath. Source data are provided with this paper. The source data underlying Figs. 1c, d, 2, 4, and 5, and Supplementary Figs. 1, 2a-d, 3, 4, 5, 6, 7b, c, and 8b-e are provided as Source Data file. Figs. 1b, 3, and Supplementary Fig. 8a are based on data represented in Supplementary Data 1 and 4. Source data are provided with this paper.

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

## Acknowledgements

We thank Raoul de Groot for insightful discussions. This work was supported by research grants from the Netherlands Organisation for Scientific Research (NWO-VENI-863.13.008 to M.A.L., NWO-ALW-ALWOP.351 to E.N.M.N't.H. and NWO-ECHO-711.017.002 to F.J.M.v.K.) and NIH (grant R01AI125561 to G.A.B.). P.G., H.P. and A.J.R. H. acknowledge support from NWO through the large-scale roadmap facility funding Proteins@Work (project 184.032.201) and X-Omics (project 184.034.019) embedded in The Netherlands Proteomics Centre. The collaboration between the groups of A.J.R.H. and F.J.M.v.K. is further supported through the Utrecht Molecular Immunology Hub.

## Author contributions

P.G. and J.R.P.M.S. contributed equally. K.A.Y.D. and I.C. contributed equally. A.J.R.H. and F.J.M.v.K. contributed equally. P.G. conceived the project, designed experiments, performed experiments, analysed data, wrote the paper. J.R.P.M.S. conceived the project, designed experiments, performed experiments, analysed data, wrote the paper. K.A.Y.D. designed experiments, performed experiments, analysed data. I.C. performed experiments, analysed data. A.M.S.B. performed experiments, analysed data. H.P. performed experiments. V.Q.T.H. performed experiments, analysed data. E.G.V. performed experiments, analysed data. M.A.L. conceived the project, designed experiments, performed experiments, provided funding, analysed data. G.A.B. designed experiments, analysed data, provided funding. E.N.M.N't.H. designed experiments, analysed data, provided funding. A.J.R.H. conceived and supervised the project, provided funding, wrote the paper. F.J.M.v.K. conceived and supervised the project, provided funding, wrote the paper.

## Competing interests

The authors declare no competing interests.
