## [Peer Review File · Nature Communications]

Reviewers' comments:

Reviewer #1 (Remarks to the Author):

Giansanti et al. - Dynamic remodeling of the human host cell (phospho)proteome upon enterovirus infection

GENERAL COMMENTS

This manuscript explores the altered proteome in cells infected with Coxsackievirus B3. The authors use MS proteomics to assess the change in the proteome and phosphoproteome in infected cells. They find phosphoproteomic changes early in infection, while proteomic changes are only detected in later timepoints (after ~8 hrs). The authors use the patterns of phosphorylation to predict the kinases involved and investigate their effects on the mTOR pathway specifically. Through investigation of the downstream targets of mTOR, they show that eEF2 phosphorylation is decreased over the course of infection. The authors also discuss the role of mTOR target TFEB, a regulator of lysosomal and autophagy. They find that TFEB is activated by infection and that knockdown of this factor inhibits virus production, but not replication, suggesting a role for lysosomal/autophagic pathways in egress.

The work reported in the manuscript is well done and describes the phosphoproteomic flux associated with infection, which is sure to be a useful reference and source for hypotheses for others in the field. The paper itself focuses on mTOR and its downstream targets. These are known to be involved in enterovirus biology, as the authors note, which makes their observation less novel, but still provides more context for mTOR involvement. The authors suggest one downstream target (TFEB) of mTOR that effects virus production, but again this observation lacks mechanistic insight beyond the genetic experiment.

Because of its quality, I believe the paper does merit acceptance and publication after the authors address some of the comments below.

Specific Comments:

II. 152-5. "While a decrease in protein levels may be explained by the virus-induced shutdown of protein translation, the observation that certain protein

154 levels increased suggests that a subset of cellular proteins escape the bulk translational shutdown induced by viral cleavage of eIF4G." Can the author distinguish between altered turnover vs. EIF4G independent translation? Any analysis that could be performed to confirm this hypothesis?

Supp. Table 4.3: Why do the authors include Apoptotic signaling in the translation and transcription? Perhaps labeling the top enriched categories on the plots would be useful for transparency.

Have the authors considered a gene set enrichment approach based on fold-change of factors in given pathways. e.g. Do genes in this pathway have a greater fold-change than expected by chance, this might better reflect the phenotype of the infected cells.

Small Notes:

I recommend renaming the section called "Regulation of viral protein levels during CVB3 expression" as the section does not discuss regulation per se.

Coxsackievirus is capitalized as it is named after a city "Coxsackie, NY, USA"

Figures: The Blue gradient background is distracting and unnecessary, makes the grey lines and blue boxes hard to visualize.

I'd encourage the authors to include some of the more "systems level" figures in the main text to give a general sense of the hits on the screen (S2C specifically).

Reviewer #2 (Remarks to the Author):

In the manuscript "Dynamic remodeling of the human host cell (phospho)proteome Upon enterovirus infection" Giansanti et al present a comprehensive proteomic and phosphoproteomic dataset across a time course of enterovirus infection. The extensive datasets are analyzed to identify

changes on the protein and phosphorylation levels and described globally. To interpret findings in the phosphoproteomic dataset further analysis was performed to predict kinases that are active during enterovirus infection. The authors then pick the mTORC1 signaling pathways for more in depth description of its possible regulation during infection and provide limited follow up studies, which do not lead to conclusive biological insights in enterovirus infection.

My main concerns regarding the current version of the manuscript are as follows: While the manuscript describes an extensive dataset that seems to be of high quality, the manuscript is mainly focused on describing the dataset rather than deriving biological insights into enterovirus infection and identifying novel host factors/signaling pathways relevant for infection. The majority of the paragraphs are highly descriptive, but the observations are not really linked back to the biological question. The authors derive some hypotheses from the dataset, but they either do not provide any functional follow up study to validate the hypothesis and the follow up studies they performed do not provide conclusive results.

Some concerns/questions that I have regarding the proteomic datasets:

Currently the authors analyze the proteome and phosphoproteome datasets separately. It would be important to determine whether observed changes in phosphorylation are in fact changing because of different levels of phosphorylation or because of protein abundance changes. This is especially important at later stages of infection where more changes on the proteome level are observed.

Some of the earlier time points for the proteome analysis do not cluster at all with the replicates. Is this a reason why there are little significant protein abundance changes observed at earlier timepoints of infection? Were the proteome and phosphoproteome datasets generated from the same samples?

In summary, the current version of the manuscript lacks novel biological insights into CVBB3 infection and revisions are necessary to (1) put results in a biological context rather than just providing a detailed description, (2) provide additional functional follow up studies demonstrating the validity of the datasets, and (3) address the specific concerns listed below.

Specific concerns:

Page 4, Introduction

The authors should provide a few sentence summary of the major findings of their study at the end of the introduction. What was the specific goal/motivation of the study? What was the motivation to use CVB3?

Page 5, Results, Analysis of the proteome of infected cells

- What is the biological motivation to use CVB3 in the study? Why was an MOI of 10 used and what was the infection rate obtained with this MOI? How reproducible was the infection?
- Reproducibility of the infection seem concerning, since the clustering of the overall proteome data is variable and shows that replicates of earlier timepoint of the infection do not cluster at all with their respective biological replicates.
- How were the clusters defined in Figure 2a (no dendogram provided)?
- Sections of this paragraphs sound just like a description of the method and could be shortened.
- I would suggest to restructure the manuscript. The flow of the manuscript feels interrupted, since the overall proteome dataset is descried in the first paragraph, but then not followed by the analysis of the proteome changes in the course of infection.

Page 6, Results, Analysis of the phosphoproteome of infected cells

- See above, sections of this paragraphs sound just like a description of the method and could be shortened.
- How were the clusters defined in Figure 2b (no dendogram provided)?
- It seems surprising to me that 85% of the phosphoproteome is changing in the course of infection. How does this compare to other phosphoproteomic datasets generated in the course of viral infection? How was the data normalized? Often in proteomic datasets normalization strategies are employed which assume that the majority of the sites are not changing. How did the authors address this in their data analysis? More details need to be provided in the method section?
- See above, the manuscript feels interrupted, since the description of the phosphoproteomic dataset is not followed by the analysis of the phosphorylation changes.

Page 6, Results, Regulation of viral protein levels during CVB3 infection

- This paragraph is purely descriptive and out of the context in this part of the manuscript.
- I would suggest to link it ack to CVB3 biology, otherwise I would remove it completely.

Page 7, Results, Changes in host protein levels during CVB3 infection

- I think for a better reading of the manuscript the authors should consider to restructure the result section and move this section after describing the proteome dataset.

- See above, it is concerning to me that replicates of some timepoints do not cluster at all and look very different in the heatmap. What is the reason for this? Was there a quality control for the reproducibility of infection?
- Paragraph misses more biology, what are these proteins that escape virus-induced translational shutdown?
- It would also be highly interesting if the authors could perform an integrative analysis of their data with the CVB3 host-pathogen interaction network that was recently published in Nature Microbiology.

Page 8, Results, Regulation of protein phosphorylation levels during CVB3 infection

- I think for a better reading of the manuscript the authors should consider to restructure the result section and move this section after describing the phosphoproteome dataset.
- At the end of the section, changes in phosphoproteome and total proteome are compared, I am wondering what the fraction of the phosphoproteome changes can in fact be explained by protein abundance changes. What is the fraction of phosphosites that change while there is no change observed on the protein level?

Page 8, Results, Identification of kinases that are activated by CVB3 infection

It is stated in the text that three findings of this analysis are of particular interest, such as overrepresentation of ATM/ATR motifs. Why are these findings of particular of particular interest? What does this mean for the infection process? What is the role of ATM/ATR or c PKC and GRKs in CVB3 infection?

Page 9, Results, Phosphorylation of host factors involved in formation of viral replication organelles

The authors describe in this section the phosphorylation of know proteins that might play a role in replication organelle formation and suggest that the specific sites might be relevant for the organelle formation, but again this is only descriptive and no biological follow up is provided. What is the possible impact of knowing the phosphorylation sites of these specific factors? Does mutating these sides lead to a reduction in viral replication?

Page 11, Results, Signaling pathways affected during CVB3 infection

More details need to be provided how the pathway analysis was performed. What are the predefined pathways known to be affected by CVB3? Where are they taken from? Was the

proteome and phosphoproteome data just mapped onto pathways or was an enrichment analysis performed? Which input data was provided in for the pathway analysis, quantitative data across all time points?

Page 11, Results, Inactivation of mTORC1 signaling during CVB3 infection

Figure 3 associated with this section is very difficult to read:

I would suggest now to use the blue background, which reduces the contrast in the figure. +/- symbols are so small that they are hard to see. Also it is not clear what the different colored arrows and lines mean.

Why was a higher MOI of 50 used for validating the phosphoproteomic data by Western Blot, which itself was generated at an MOI of 10.

Page 14, Results, Activation of the mTORC1-regulated protein transcription factor EB

How was the knockout of TFEB performed? There is no information provided in the method section. Additionally, validation of the ko is missing.

What is the explanation that 8hpi a significant decrease in virus titers is observed upon ko of TFEB, however there is no significant difference 10hpi? Does this makes sense?

Page 18, Experimental procedures

As mentioned in different comments above, a more detailed description of some of the methods is missing and should be provided.

Reviewers' comments:

Reviewer #1 (Remarks to the Author):

GENERAL COMMENTS

This manuscript explores the altered proteome in cells infected with Coxsackievirus B3. The authors use MS proteomics to assess the change in the proteome and phosphoproteome in infected cells. They find phosphoproteomic changes early in infection, while proteomic changes are only detected in later timepoints (after ~8 hrs). The authors use the patterns of phosphorylation to predict the kinases involved and investigate their effects on the mTOR pathway specifically. Through investigation of the downstream targets of mTOR, they show that eEF2 phosphorylation is decreased over the course of infection. The authors also discuss the role of mTOR target TFEB, a regulator of lysosomal and autophagy. They find that TFEB is activated by infection and that knockdown of this factor inhibits virus production, but not replication, suggesting a role for lysosomal/autophagic pathways in egress. The work reported in the manuscript is well done and describes the phosphoproteomic flux associated with infection, which is sure to be a useful reference and source for hypotheses for others in the field. The paper itself focuses on mTOR and its downstream targets. These are known to be involved in enterovirus biology, as the authors note, which makes their observation less novel, but still provides more context for mTOR involvement. The authors suggest one downstream target (TFEB) of mTOR that effects virus production, but again this observation lacks mechanistic insight beyond the genetic experiment. Because of its quality, I believe the paper does merit acceptance and publication after the authors address some of the comments below.

Specific Comments:

“While a decrease in protein levels may be explained by the virus-induced shutdown of protein translation, the observation that certain protein levels increased suggests that a subset of cellular proteins escape the bulk translational shutdown induced by viral cleavage of eIF4G.” Can the author distinguish between altered turnover vs. EIF4G independent translation? Any analysis that could be performed to confirm this hypothesis?

It is difficult to distinguish between altered turnover vs eIF4G-independent translation. We believe that the most direct way to gain some insight into this would be to perform experiments with virus mutants that have a reduced ability to cleave eIF4G or under conditions where protein turnover is affected. However, both manipulations will affect the overall efficiency and dynamics of virus replication. Mutations in the viral protease 2A^{pro} that affect its trans-cleavage activity will not only affect eIF4G cleavage but also the 2A^{pro}-mediated cleavage of other host factors. Alternatively cellular protein turn-over can be manipulated (e.g. by proteasome inhibition), but this has been reported to inhibit CVB3 replication, thereby also hampering 2A^{pro} expression and eIF4G cleavage. Thus, all in all, we think that within the limits of current available methodology no analysis or experiment can be done to answer this interesting question unambiguously.

Supp. Table 4.3: Why do the authors include Apoptotic signaling in the translation and transcription? Perhaps labeling the top enriched categories on the plots would be useful for transparency.

We thank the reviewer for noting this and need to apologize for this mistake, which has now been corrected. This table, which has become Supplemental Table S2.3 in the revised version of our manuscript contains the data used for Supplemental Figures S2a and S2b. In the plots, we chose to highlight particularly clusters that contain the most GO categories and clusters being most relevant for CVB3 infection, which especially for the proteome encompass also the most enriched categories.

We now also added Supplemental Table S2.4, which describes the PhosphoPath analysis of the proteome and phosphoproteome data that is displayed in Supplemental Figure S2d.

Have the authors considered a gene set enrichment approach based on fold-change of factors in given pathways. e.g. Do genes in this pathway have a greater fold-change than expected by chance, this might better reflect the phenotype of the infected cells.

A gene set enrichment analysis (GSEA) requires a rather complete and homogeneous coverage of all the enriched pathways, which unfortunately is often not the case in MS-based proteomics studies. Instead, we used the PhosphoPath workflow, which nicely allowed us to integrate the proteome and phosphoproteome data. A GSEA would have to be performed on the combined proteomics and phosphoproteomics datasets, which to the best of our knowledge is not possible in any of the currently available software packages.

Small Notes:

I recommend renaming the section called “Regulation of viral protein levels during CVB3 expression” as the section does not discuss regulation perse.

In relation also to a comment of reviewer #2, we have moved a shortened version of this section to end of the first Results section (section “Analysis of the proteome of infected cells”) (page 6 lines 138-145 of the revised manuscript).

Coxsackievirus is capitalized as it is named after a city “Coxsackie, NY, USA”

Indeed, coxsackieviruses are named after the city Coxsackie, NY, USA. Historically, the name of the viruses has therefore often been capitalized. Recently, the Picornavirus Study Group of the International Committee for the Taxonomy of Viruses (ICTV) has recommended not to capitalize these names (Simmonds P, et al., 2020, Arch Virol; <https://doi.org/10.1007/s00705-019-04520-6>). We have decided to follow this recommendation.

Figures: The Blue gradient background is distracting and unnecessary, makes the grey lines and blue boxes hard to visualize.

Thanks for this note. We have modified the figure as requested.

I'd encourage the authors to include some of the more “systems level” figures in the main text to give a general sense of the hits on the screen (S2C specifically).

We expect the reviewer is referring to Figure S3c, which has become S2b in the revised version. We have chosen to include this figure in the Supplement given its size (the figure will not meet the journal guidelines in terms of font size and lines width) and because it is not of core importance for the main flow of the manuscript. We can convert it into a separate main figure if the editor/journal would request so.

Reviewer #2 (Remarks to the Author):

In the manuscript “Dynamic remodeling of the human host cell (phospho)proteome Upon enterovirus infection” Giansanti et al present a comprehensive proteomic and phosphoproteomic dataset across a time course of enterovirus infection. The extensive datasets are analyzed to identify changes on the protein and phosphorylation levels and described globally. To interpret findings in the phosphoproteomic dataset further analysis was performed to predict kinases that are active during enterovirus infection. The authors then pick the mTORC1 signaling pathways for more in depth description of its possible regulation during infection and provide limited follow up studies, which do not lead to conclusive biological insights in enterovirus infection.

My main concerns regarding the current version of the manuscript are as follows: While the manuscript describes an extensive dataset that seems to be of high quality, the manuscript is mainly focused on describing the dataset rather than deriving biological insights into enterovirus infection and identifying novel host factors/signaling pathways relevant for infection. The majority of the paragraphs are highly descriptive, but the observations are not really linked back to the biological question. The authors derive some hypotheses from the dataset, but they either do not provide any functional follow up study to validate the hypothesis and the follow up studies they performed do not provide conclusive results.

We thank the reviewer for his appreciation of the high-quality of our datasets. In our system-wide approach, whereby we observed so many phosphosites and proteins to be regulated, there are (too) many leads to follow up, and it is hard to make a judgment which one will be most meaningful. Although we described already quite a few validation experiments in the initial manuscript, we have now performed extensive new experiments and provide especially new insights into the role of TFEB in secretory autophagy during CVB3 infection (Figures 5b-e, Supplemental Figure S7 and page 13 line 363 – page 14 line 380 of the revised manuscript) as well as on the role of GBF1 and PI4KB phosphorylation in the formation of the replication organelles (Supplemental Figures S8c-e, page 14 line 400 – page 15 line 409 and page 31-32 Supplemental text of the revised manuscript). We expect that these new data are resolving the reviewers issue.

Some concerns/questions that I have regarding the proteomic datasets:

Currently the authors analyze the proteome and phosphoproteome datasets separately. It would be important to determine whether observed changes in phosphorylation are in fact changing because of different levels of phosphorylation or because of protein abundance changes. This is especially important at later stages of infection where more changes on the proteome level are observed. Some of the earlier time points for the proteome analysis do not cluster at all with the replicates. Is this a reason why there are

little significant protein abundance changes observed at earlier timepoints of infection? Were the proteome and phosphoproteome datasets generated from the same samples?

We have addressed these concerns point-by-point below.

In summary, the current version of the manuscript lacks novel biological insights into CVB3 infection and revisions are necessary to (1) put results in a biological context rather than just providing a detailed description, (2) provide additional functional follow up studies demonstrating the validity of the datasets, and (3) address the specific concerns listed below.

As described above, we have now add new data that provide new insights into the role of TFEB during CVB3 infection as well as in the role of GBF1 and PI4KB phosphorylation for replication organelle formation.

Specific concerns:

Page 4, Introduction

The authors should provide a few sentence summary of the major findings of their study at the end of the introduction. What was the specific goal/motivation of the study? What was the motivation to use CVB3?

As suggested, we have added a few sentences to motivate our study and to summarize the main outcome (page 4 lines 84-93 of the revised manuscript).

Page 5, Results, Analysis of the proteome of infected cells

- What is the biological motivation to use CVB3 in the study? Why was an MOI of 10 used and what was the infection rate obtained with this MOI? How reproducible was the infection?

We used CVB3 because this is one the model viruses most commonly used nowadays in enterovirus research. This virus is also the model virus that we have already studied in detail for over 25 years in our lab. Data obtained with this virus can thus be nicely connected to previously obtained insights into the biology of this virus. Importantly, many aspects of infection are similar between different enteroviruses, implying a wider impact of our findings in the field. Indeed, we show that a key finding of our study, namely mTOR inactivation during infection, also occurs during infection with poliovirus (which belongs to a different enterovirus species than CVB3).

We used an MOI of 10 to ensure high infection efficiency so that all cells are infected. Infections at this MOI are commonly used in our lab because infections are efficient and because viral replication kinetics and viral effects on host cell processes are highly reproducible.

- Reproducibility of the infection seem concerning, since the clustering of the overall proteome data is variable and shows that replicates of earlier timepoint of the infection do not cluster at all with their respective biological replicates.

Clustering is relatively poor only for some samples in the proteome dataset, especially at early time points. Two effects are occurring that may cause this. First, changes in the proteome in general are much smaller

than in the phosphoproteome. As a result, small variations between samples are relatively big as compared to the overall proteome change and affect clustering relatively strongly (which is not seen in the phosphoproteome). Second, especially at early time points changes are even smaller and hence any variation between samples may appear even bigger and lead to poorer clustering. Indeed, clustering of proteome samples from 6 hpi onward (and from controls) is much better than at earlier time points.

- How were the clusters defined in Figure 2a (no dendogram provided)?

We have modified Figure 2a to include the rows dendogram and revised the legend and the Experimental Procedures subsection 'Data analysis of (phospho)proteomics' to provide more information on the cluster analysis.

- Sections of this paragraphs sound just like a description of the method and could be shortened.

As suggested, we shortened this paragraph wherever possible to prevent too much overlap with description of the method while retaining details that we believe are required for the readership to easily read and understand the results.

- I would suggest to restructure the manuscript. The flow of the manuscript feels interrupted, since the overall proteome dataset is described in the first paragraph, but then not followed by the analysis of the proteome changes in the course of infection.

We have restructured the manuscript as requested.

Page 6, Results, Analysis of the phosphoproteome of infected cells

- See above, sections of this paragraphs sound just like a description of the method and could be shortened.

As suggested, we shortened this paragraph wherever possible to prevent too much overlap with description of the method while retaining details that we believe are required for the readership to easily read and understand the results.

- How were the clusters defined in Figure 2b (no dendogram provided)?

In response, Figure 2b has been modified to include the rows dendogram and we revised the legend and the Experimental Procedures subsection 'Data analysis of (phospho)proteomics' (page 20-21) to provide more information on the cluster analysis.

- It seems surprising to me that 85% of the phosphoproteome is changing in the course of infection. How does this compare to other phosphoproteomic datasets generated in the course of viral infection? How was the data normalized? Often in proteomic datasets normalization strategies are employed which assume that the majority of the sites are not changing. How did the authors address this in their data analysis? More details need to be provided in the method section?

We apologize for the confusion. The number 85% indicates the quantified phosphoproteome (i.e. those sites quantified in all biological replica in at least one time point, see revised 'Data analysis of (phospho)proteomics' section) that is sensitive to infection. Similarly, the number for the proteome concerns only the quantified proteome. This does not imply that 85% of the entire cellular phosphoproteome changes altogether throughout the course of the infection. As can be seen particularly in Figure 2b, sites undergo differential regulation. The 85% hence indicates the sum of all quantified phosphosites that are changed at any time point divided by the total number of quantified phosphosites. We have clarified this in the text now (page 7 lines 163-1166 of the revised manuscript).

Furthermore, it is important to stress that - as can be seen in Figure 2b, Supplemental Figure 1c (right) and Supplemental Table S4 - regulations happen in multiple waves. A small fraction of sites is regulated at early time points (2 or 4 hpi, ~15%), other at intermediate times (6 hpi, ~30%) and substantial fraction at the latest time points (8 or 10 hpi, ~50%).

To the best of our knowledge, our work represents the first large-scale (phospho)proteomics analysis of the host response upon enterovirus infection. We believe that a comparison of our data against data generated in the course of infection with other viruses will unlikely bring meaningful information, as different viruses replicate via different mechanisms and target different cellular pathways, resulting in different host phosphoproteome outcomes. Nevertheless, we looked at two other phosphoproteomics datasets obtained upon virus infection. Söderholm *et al* (<https://doi.org/10.1074/mcp.M116.057984>) reported that ~26% of the human macrophages phosphoproteome changes in response to influenza A virus infection, whereas Mohl *et al* (<https://doi.org/10.1074/mcp.M117.067355>) reported that ~53% of the HeLa phosphoproteome undergoes regulation upon Bluetongue virus infection. Therefore, the numbers we report are high, but not unexpected.

The MS data were normalized by median-centering the protein or site intensity within each sample, a well-accepted and commonly used normalization strategy in the MS-based proteomics. We revised the 'Data analysis of (phospho)proteomics' section and added more detailed descriptions for analysis of the MS data, including filtering and normalization applied (page 21 lines 562-565 of the revised manuscript).

- See above, the manuscript feels interrupted, since the description of the phosphoproteomic dataset is not followed by the analysis of the phosphorylation changes.

As requested, we have now restructured the manuscript.

Page 6, Results, Regulation of viral protein levels during CVB3 infection

- This paragraph is purely descriptive and out of the context in this part of the manuscript.
- I would suggest to link it back to CVB3 biology, otherwise I would remove it completely.

We put in the paragraph about the viral proteins primarily to show how fast the virus takes over the host replication machinery, which we think also further confirms that virus infection was successful and reasonably reproducible. Therefore, we suggest to keep the information in, although we have shortened

the section and moved it the first paragraph of the Results section (“Analysis of the proteome of infected cells”) (page 6 lines 138-145 of the revised manuscript).

Page 7, Results, Changes in host protein levels during CVB3 infection

- I think for a better reading of the manuscript the authors should consider to restructure the result section and move this section after describing the proteome dataset.

As requested, we have restructured the manuscript, shortened this section and moved it to the end of the Results section (page 14 line 400). Additionally, we have performed additional experiments (Supplemental Figure S8c-e) that are now described in Supplementary text (page 31 lines 887 – page 32 line 911 of the revised manuscript).

- See above, it is concerning to me that replicates of some timepoints do not cluster at all and look very different in the heatmap. What is the reason for this? Was there a quality control for the reproducibility of infection?

As mentioned above, clustering is only poor at early time points where the detected changes are extremely small. For the proteome data, that is certainly true for Mock, 2 hpi and 4 hpi, implying that there are only minor changes at these time points as compared to the control. Obviously, this is mainly because at this early time point the infection is just kicking off and thereby (small) variations between samples will look more prominent. Of note, clustering is very good at all later time points. For the phosphoproteome, only the two controls and the 2 hpi samples mix up in the clusters for the same reason. Phosphoproteome changes are much more extensive and quicker than proteome changes and therefore the clustering is already very good from 4 hpi onward.

- Paragraph misses more biology, what are these proteins that escape virus-induced translational shutdown?

We now more explicitly mention in the text (page 5 line 124 of the revised manuscript) that two groups of proteins stand out, namely a subset of mitochondrial proteins and several lysosomal proteins. The lysosomal proteins are discussed extensively (at a later point) in the manuscript, as they are downstream of mTORC1 signaling and subject to regulation through TFEB.

- It would also be highly interesting if the authors could perform an integrative analysis of their data with the CVB3 host-pathogen interaction network that was recently published in Nature Microbiology.

The paper that the reviewer presumably refers to is “Diep J *et al* (2019). Enterovirus pathogenesis requires the host methyltransferase SETD3. *Nat Microbiol* 4: 2523-2537. In this paper, the authors describe a genome-wide CRISPR screen to identify host factors involved in enterovirus replication. They used affinity-purification mass spectrometry to identify host factors interacting with viral proteins.

The approach by Diep J *et al* and the generated dataset is quite different than the approach we apply here. Unfortunately, we think that comparing these two datasets will not yield a lot of meaningful insights.

First, changes in host cell protein levels as monitored by us likely occur independent of the interactions between viral and host cellular proteins monitored by Diep *et al.* Second, as enteroviruses do not contain any proteins with kinase or phosphatase activity, the monitored changes in the phosphoproteome are highly unlikely to be related to interactions between viral and host cellular proteins.

Page 8, Results, Regulation of protein phosphorylation levels during CVB3 infection

- I think for a better reading of the manuscript the authors should consider to restructure the result section and move this section after describing the phosphoproteome dataset.

As requested, we have now restructured the manuscript.

- At the end of the section, changes in phosphoproteome and total proteome are compared, I am wondering what the fraction of the phosphoproteome changes can in fact be explained by protein abundance changes. What is the fraction of phosphosites that change while there is no change observed on the protein level?

In our system wide approach, we monitor thousands of proteins and thousands of phosphosites. We do not expect that there will be a simple correlation between changes at the protein level and changes at the phosphorylation level. To simply illustrate this a given protein may harbor phosphosites that go highly up, highly down and remain unaffected. To illustrate this further on our data we plotted the fold changes (10h-vs-CT) of the proteome vs the phosphoproteome obtaining the following distribution:

Here, each point represents a phosphosite, its x coordinate represents the corresponding protein fold change, while the y coordinate represents the site fold change. There is neither a clear trend nor a good

correlation ($r = 0.12$), implying that phosphoproteome changes are largely independent from proteome changes.

Page 8, Results, Identification of kinases that are activated by CVB3 infection

It is stated in the text that three findings of this analysis are of particular interest, such as overrepresentation of ATM/ATR motifs. Why are these findings of particular of particular interest? What does this mean for the infection process? What is the role of ATM/ATR or c PKC and GRKs in CVB3 infection?

These findings stood out because they appeared in the same cluster in two different analyses (motif-X and NetworKIN) (Figure 2c and Supplemental Figure S4a). To avoid misunderstanding we rephrased the sentence and removed 'of particular interest' (page 8 lines 209-217 of the revised manuscript).

Several PKC and GRK motifs were highly enriched in cluster 2; presently, we do not know whether this has any implications for infection. The group of ATM/ATR motifs drew our particular attention because they are very highly enriched in cluster 1 and because mTORC, a member of the ATM/ATR kinase family, had previously been implicated in enterovirus infection (Supplemental Figure S2d). Of the kinases specifically mentioned, we follow up mTOR in detail. The observation that ATM/ATR sites decrease in phosphorylation aligns with our finding that mTORC1 activity decreases as infection progresses.

Page 9, Results, Phosphorylation of host factors involved in formation of viral replication organelles

The authors describe in this section the phosphorylation of known proteins that might play a role in replication organelle formation and suggest that the specific sites might be relevant for the organelle formation, but again this is only descriptive and no biological follow up is provided. What is the possible impact of knowing the phosphorylation sites of these specific factors? Does mutating these sites lead to a reduction in viral replication?

In response to the reviewer, we added new data (Supplemental Figure S8c-e) in which we tested whether a number of phosphosite mutants of GBF1 or PI4KB could support replication in experiment systems in which the endogenous proteins are functionally nullified.

Page 11, Results, Signaling pathways affected during CVB3 infection

More details need to be provided how the pathway analysis was performed. What are the predefined pathways known to be affected by CVB3? Where are they taken from? Was the proteome and phosphoproteome data just mapped onto pathways or was an enrichment analysis performed? Which input data was provided in for the pathway analysis, quantitative data across all time points?

We apologize for the lack of information. The input for PhosphoPath analysis were the quantitative proteome and phosphoproteome data across all time points. We have revised the text (page 9 lines 224-228 of the revised manuscript). We have also revised the 'Data analysis of (phospho)proteomics' section in the Experimental procedures section to add additional information about PhosphoPath (page 21 lines 584-588 of the revised manuscript).

Regarding the predefined pathways known to be affected by CVB3, they were selected upon extensive literature mining. The effect of CVB3 on these pathways is described in more details in Si X *et al*, Esfandiarei M & McManus B, and Chang H *et al*, already referenced in the original version of our manuscript (Ref. 23-25 of the original manuscript).

Page 11, Results, Inactivation of mTORC1 signaling during CVB3 infection

Figure 3 associated with this section is very difficult to read:

I would suggest now to use the blue background, which reduces the contrast in the figure. +/- symbols are so small that they are hard to see. Also it is not clear what the different colored arrows and lines mean.

We have modified the figure as requested by both reviewers.

Why was a higher MOI of 50 used for validating the phosphoproteomic data by Western Blot, which itself was generated at an MOI of 10.

At this higher MOI, the effects of viral infection on cellular proteins are more clearly visible. But similar effects can be observed at MOI of 10. Thus, the overall outcome is not different.

Page 14, Results, Activation of the mTORC1-regulated protein transcription factor EB

How was the knockout of TFEB performed? There is no information provided in the method section. Additionally, validation of the ko is missing.

The HAP1 TFEB knockout cells were obtained from a commercial source, as described in the Experimental procedures section. Details on how the gene was knocked out by CRISPR are available on the supplier's website. We confirmed the deletion in the TFEB gene by sequencing using the same procedures as used by the supplier to characterize the cells, which we have now mentioned in the Experimental procedures section (page 18 lines 486-489 of the revised manuscript).

What is the explanation that 8hpi a significant decrease in virus titers is observed upon ko of TFEB, however there is no significant difference 10hpi? Does this makes sense?

We now added new data to further investigate this phenomenon (Figure 5, Supplemental Figure S7, and page 13 line 363 - page 14 line 390 of the revised manuscript). Briefly, we show that the difference in virus release is not related to any difference in cell integrity. Instead, we show a clear reduction in both free extracellular virus and virus in extracellular vesicles. In line with this, we show that the release of LC3-positive extracellular vesicles, which are generated in an autophagic process, is essentially aborted in TFEB KO cells. Hence, the decreased extracellular titers in the TFEB^{KO} condition can be explained by a marked reduction of TFEB-dependent secretory autophagy. At 10 hpi, cells have gone into demise and lytic release has become the major source of extracellular virus and any difference in extracellular virus in vesicles – if there is any – is masked by the vast excess of free extracellular virus.

Page 18, Experimental procedures

As mentioned in different comments above, a more detailed description of some of the methods is missing and should be provided.

To improve clarity, we have now provided additional details on the methods used as mentioned above.

REVIEWERS' COMMENTS:

Reviewer #1 (Remarks to the Author):

The revised version of this manuscript address all the questions and suggestions by the authors and in my opinion the majority of the concerns describe by the other reviewer. I believe that the inclusion of of additional experimental validations on GBF1 and PI4KB phosphorylation in the formation of the replication organelles and the connection of TFEB and secretory autophagy during CVB3 infection is interesting finding. I believe the paper has significantly increased in quality and I have not additional comments or concerns.

Reviewer #2 (Remarks to the Author):

For the revised version of the manuscript entitled “Dynamic remodeling of the host cell (phospho)proteome upon enterovirus infection” the authors addressed the majority of my comments. The main concerns that I raised included the focus on the previous version on the description of the dataset rather than emphasizing what the data tells us about the biology of enterovirus infection, the lack of functional follow up studies, and restructuring of the manuscript.

The authors restructured the manuscript substantially as well as shorten some of the descriptive sections, so that it is more intuitive for the reader to follow the major findings of the study.

In order to address the lack of biological insights extracted from the large scale dataset the authors added functional data characterizing the role of mTORC1 inactivation leading to increased TFEB activity as well as investigating the role of phosphorylation of host factors involved in the formation of viral replication organelles.

I have a few remaining concerns/questions regarding the functional follow up findings:

1. The authors hypothesize that TFEB activation downstream of mTORC1 affect non-lytic virus release vis extracellular vesicles (EV). While the authors show a significantly reduced EV-enclosed virus release, but at the same time TFEB ko also reduces the naked virus, even though it is not significant reduced, which is likely due to variability in the measurements/experiment. Does this result imply that TFEB does not only affect the non-lytic but also the lytic virus release? How do the authors explain ths obersevation?

2. The last sentence of the Result section “These data lend further support to the idea that (entero)viruses not only hijack cellular pathways to build their replication organelles but that they

also rewire them from their regulation in non-infected cells.” seems unclear to me. Please clarify the meaning of “...rewire them from their regulation in non-infected cells.”

Reviewers' comments:

Reviewer #2:

1. The authors hypothesize that TFEB activation downstream of mTORC1 affect non-lytic virus release via extracellular vesicles (EV). While the authors show a significantly reduced EV-enclosed virus release, but at the same time TFEB ko also reduces the naked virus, even though it is not significant reduced, which is likely due to variability in the measurements/experiment. Does this result imply that TFEB does not only affect the non-lytic but also the lytic virus release? How do the authors explain the observation?

The referee is correct that we observed a small, but not significant, reduction in release of naked virus. This is indeed likely due to variations in the experiments, but we cannot rule out some contribution of TFEB to lytic virus release. So therefore, following the sentence "In addition, a trend for a reduction in the release of naked virus particles was observed.", we now added the sentence "This reduction may be due to variability in the experiments although we cannot formally rule out a contribution of TFEB in lytic virus release."

2. The last sentence of the Result section "These data lend further support to the idea that (entero)viruses not only hijack cellular pathways to build their replication organelles but that they also rewire them from their regulation in non-infected cells." seems unclear to me. Please clarify the meaning of "...rewire them from their regulation in non-infected cells."

This sentence was indeed complicated and therefore we changed the text into preceding the last sentence into "As discussed more extensively in the Supplemental text and shown in Supplemental Figure S8, when the activity or expression of the endogenous proteins were disrupted, phosphomimetic and non-phosphorylatable mutants of these sites could both restore virus replication, indicating that phosphorylation at sites that have been implicated to be important for membrane homeostasis in uninfected cells are not important for enteroviruses to build their replication organelles.", and we simplified the last sentence into "These data lend further support to the idea that enteroviruses can rewire cellular pathways for efficient replication."